# SUMO2 conjugation of PCNA facilitates chromatin remodeling to resolve transcription-replication conflicts

Min Li[1], Xiaohua Xu [1], Chou-Wei Chang [1], Li Zheng[1], Binghui Shen[1] & Yilun Liu [1]

During DNA synthesis, DNA replication and transcription machinery can collide, and the replication fork may temporarily dislodge RNA polymerase II (RNAPII) to resolve the transcription-replication conflict (TRC), a major source of endogenous DNA double-strand breaks (DSBs) and common fragile site (CFS) instability. However, the mechanism of TRC resolution remains unclear. Here, we show that conjugation of SUMO2, but not SUMO1 or SUMO3, to the essential replication factor PCNA is induced on transcribed chromatin by the RNAPII-bound helicase RECQ5. Proteomic analysis reveals that SUMO2-PCNA enriches histone chaperones CAF1 and FACT in the replication complex via interactions with their SUMO-interacting motifs. SUMO2-PCNA enhances CAF1-dependent histone deposition, which correlates with increased histone H3.1 at CFSs and repressive histone marks in the chromatin to reduce chromatin accessibility. Hence, SUMO2-PCNA dislodges RNAPII at CFSs, and overexpressing either SUMO2-PCNA or CAF1 reduces the incidence of DSBs in TRC-prone RECQ5-deficient cells.

[1] Department of Cancer Genetics and Epigenetics, Beckman Research Institute, City of Hope, Duarte, CA 91010-3000, USA. These authors contributed equally: Min Li, Xiaohua Xu. Correspondence and requests for materials should be addressed to Y.L. (email: yiliu@coh.org)

D NA damaging agents can generate DNA double-strand breaks (DSBs). However, in the absence of exogenous assault during unperturbed cell growth, DSBs can also occur due to transcription-replication conflicts (TRCs). For example, the collision between RNA polymerase II (RNAPII) and a replisome can halt progression of the replication fork, leading to fork collapse, DSB formation, and genomic instability[1,2]. TRC-induced DSBs are a major cause of common fragile site (CFS) instability, which takes place within gene regions that are predominantly transcribed during S-phase[1,3–5]. CFS instability leads to genomic rearrangements, loss of heterozygosity, and microsatellite instability, all of which are drivers for cancer pathogenesis[6,7]. Indeed, several CFS-containing genes, including tumor suppressor WWOX, are among the most frequently deleted gene loci in cancer cells[8]. A possible mechanism to resolve TRC is to temporarily remove RNAPII in the collision path without dissociating the RNA transcript[1]. The displaced RNAPII may re-associate with the DNA template to resume transcription after the replication fork passes[1]. In bacteria, this removal may be achieved by DNA motor proteins that can push RNAPII off the DNA[9]. However, the mechanism by which RNAPII is dislodged in higher eukaryotes remains unclear.

Proliferating cell nuclear antigen (PCNA), an essential component of the DNA replication fork, enhances the processivity of DNA polymerases during DNA synthesis. In addition, PCNA is ubiquitinated at lysine 164 (K164) in response to DNA damage to facilitate DNA lesion bypass[10]. In unperturbed cells, K164 can also be conjugated with either small ubiquitin-like modifier 1 (SUMO1) or SUMO2[11–13]. However, although the SUMO1-PCNA conjugate has been implicated in recruiting the PCNA-associated recombination inhibitor (PARI) helicase to the replication fork to suppress unwanted homologous recombination (HR)[12,13], it is not known if SUMO2-PCNA has functions that are redundant with those of SUMO1-PCNA. Furthermore, the cellular events that trigger SUMO modification of PCNA at the replication fork are not clear.

Here, we report a novel function of SUMO2-PCNA in resolving TRC. We discovered that the conjugation of PCNA to SUMO2, but not SUMO1, is induced by transcription and is unique to the replication fork associated with the transcribed region of the chromatin. We further determined that the RNAPII-bound DNA helicase RECQ5, which suppresses transcription-associated DSBs and acts as a tumor suppressor[14–20], interacts with PCNA via a newly identified PCNA-interacting protein (PIP) motif to induce SUMO2 conjugation of PCNA. Using proteomic analysis, we found that SUMO2-PCNA specifically enriches histone chaperones CAF1 and FACT within the replication complex via their SUMO-interacting motifs (SIMs). Increasing SUMO2-PCNA levels in cells enhances deposition of CAF1-dependent histone H3.1 and establishment of repressive histone marks, which coincide with the destabilization of RNAPII from the chromatin, especially at CFSs, and enhanced replication fork progression. Through these mechanisms, overexpression of SUMO2-PCNA or CAF1 reduces incidents of DSBs in TRC-prone RECQ5-deficient cells. In summary, our observations provide mechanistic insight into how SUMO2-PCNA restricts transcription via chromatin remodeling during DNA replication to minimize genomic instability that may arise from catastrophic encounters between the replication and transcription machinery.

## Results

### Conjugation of SUMO2 to PCNA is induced by transcription.
We used our previously developed chromatin isolation protocol[21] to isolate proteins associated with transcriptionally active

chromatin (i.e., the CB:RNA+ fraction in Fig. 1a), such as hyperphosphorylated RNAPII (RNAPIIo)[22]. After isolating the CB:RNA+ fraction using cellular fractionation and RNase A treatment, we digested the remaining chromatin pellet with benzonase to remove the remaining nucleic acids. The protein fraction collected after benzonase treatment was designated CB: RNA- (Fig. 1a). As expected, most activated RNAPIIo, but not inactive, non-phosphorylated RNAPII (RNAPIIa), was found in the CB:RNA+ fraction (Fig. 1b). Lamin A, which is involved in heterochromatin formation[23], was only present in the CB:RNA− fraction (Fig. 1b). In addition, we found that the splicing factors SRSF1 and U2AF65 physically associated with RNAPII in the CB: RNA+, but not the CB:RNA-, fraction (Supplementary Fig. 1a), consistent with the coupling of transcription and splicing for active mRNA production[24].

Because both transcribed and non-transcribed chromosomes must be replicated during cell growth, DNA replication factors, such as DNA polymerase δ (Pol δ) and PCNA, were found in both CB:RNA+ and CB:RNA− fractions (Fig. 1b). As expected, the presence of these DNA replication factors on the chromatin was not inhibited after the cells were treated with 5,6-dichloro-1-beta-D-ribofuranosylbenzimidazole (DRB; Fig. 1b), which blocks the phosphorylation of RNAPII[25]. Surprisingly, we observed that a subset of PCNA (PCNA*) was post-translationally modified in CB:RNA+ fractions prepared from multiple human cell lines (Fig. 1b; Supplementary Fig. 1b). This modification, which was detected only in cells undergoing DNA synthesis (Fig. 1c, d), was dependent on transcription, as DRB treatment efficiently eliminated PCNA* from the chromatin (Fig. 1b). Because PCNA* protein levels in cells were stabilized by N-ethylmaleimide (NEM; Supplementary Fig. 1c), an inhibitor of deubiquitinases, PCNA* was likely conjugated to a ubiquitin (Ub) or ubiquitin-like protein. To test this, we exogenously expressed and purified His-and Myc-tagged PCNA (His-Myc-PCNA) using Ni-NTA under denaturing conditions and confirmed that His-Myc-PCNA was also modified in the CB:RNA+ fraction (Fig. 1e). However, the purified PCNA* proteins were not recognized by an α-Ub antibody, and the PCNA* level was not enhanced after exposure to ultraviolet (UV) light, suggesting that the PCNA* modification is distinct from UV-induced PCNA ubiquitination (Supplementary Fig. 1d).

To determine the nature of the PCNA* modification, we separated the His-Myc-PCNA from a CB:RNA+ fraction purified under denaturing conditions by SDS polyacrylamide gel electrophoresis (SDS–PAGE) and excised the protein band corresponding to His-Myc-PCNA* (Supplementary Fig. 2a). Mass spectrometry analysis identified the presence of the peptide sequence "VAGQDGSVVQFK" unique to SUMO2/3, but not SUMO1, in the purified His-Myc-PCNA* sample (Supplementary Fig. 2b), suggesting that PCNA* was likely conjugated to SUMO2/3. Indeed, an α-SUMO2/3 antibody, but not one against SUMO1, recognized the His-Myc-PCNA* from the CB:RNA+ fraction (Fig. 1e; Supplementary Fig. 2c). To further distinguish between SUMO2 and SUMO3, we exogenously expressed His-tagged SUMO1, SUMO2, or SUMO3 in HEK293T cells and purified the various His-tagged SUMOs and SUMO-conjugated proteins under denaturing conditions from whole cell extracts (WCE) and CB:RNA+ fractions. Consistent with previous reports[11–13], we found that PCNA was conjugated with either SUMO1 or SUMO2, but not SUMO3, in WCE (Fig. 1f, middle panel). However, only His-SUMO2-PCNA was present in the CB:RNA+ fraction (Fig. 1f, bottom panel). Using mutagenesis analysis, we identified K164 as the SUMO2-modified residue on PCNA (Fig. 1g). Collectively, these results indicate that a subset of PCNA on transcriptionally active chromatin is SUMO2 conjugated at K164 during

S-phase in a manner dependent on RNAPII-mediated transcription.

**SUMO2-PCNA increases replication fork progression rate**. Because SUMO2-PCNA was only present in cells undergoing DNA synthesis (Fig. 1d), we wished to determine if SUMO2-

PCNA affects replication fork progression. To do this, we adopted a strategy based on previous observations that a PCNA molecule fused with Ub at its N- or C-terminus structurally resembles K164-ubiquitinated PCNA[26]. This Ub-PCNA fusion protein can function during DNA translesion synthesis to bypass DNA damage in human cells, even when the K164R (KR)

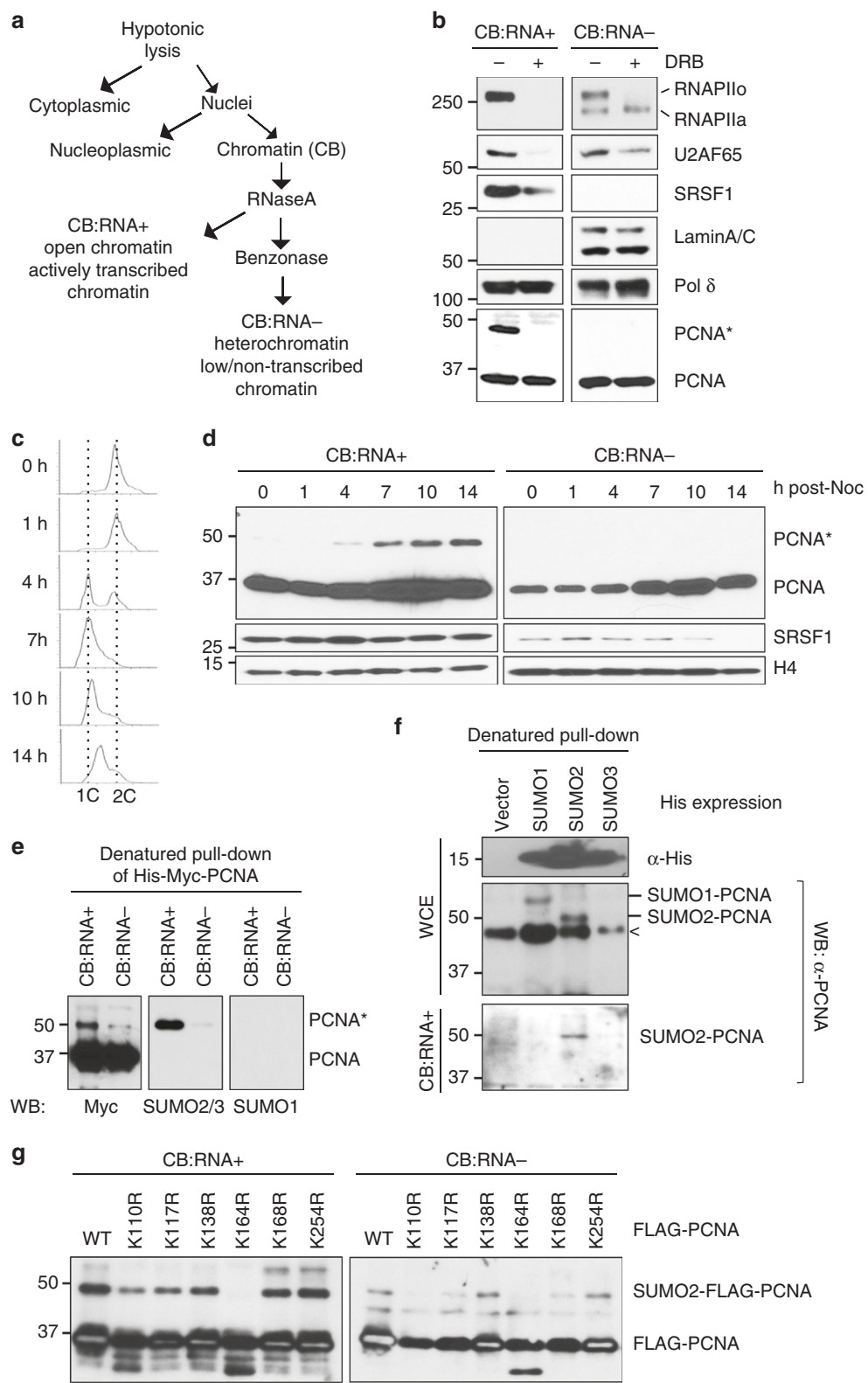

mutation is introduced and thus K164 cannot be modified[27]. Similarly, a SUMO1-PCNA (KR) fusion protein mimics the function of PCNA conjugated with a SUMO1 at K164 to suppress HR in human cells[13]. We created a SUMO2-PCNA fusion protein with the K164R mutation (S2-KR) to prevent any additional modification at this site (Fig. 2a).

Our initial attempt to generate PCNA shRNA-knockdown cells stably expressing a shRNA-resistant S2-KR fusion construct failed due to cell growth inhibition a week after transfection. Therefore, we transiently overexpressed a FLAG-tagged S2-KR fusion protein in HEK293T cells (Fig. 2b) and looked for potential dominant phenotypes caused by the overexpression of the S2-KR as compared to overexpression of wild type (WT) or KR mutant PCNA. Both the overexpressed FLAG-PCNA and endogenous PCNA proteins were found in the chromatin-bound (CB) fractions (Fig. 2c). Furthermore, PCNA overexpression did not significantly alter levels of replication factors, such as MCM2-7 (represented by MCM2) or GINS (represented by SLD5), bound to the chromatin (Fig. 2c). However, interestingly, DNA fiber analysis revealed that overexpression of WT or KR PCNA increased the average replication fork progression rate ~1.5-fold compared to transfection with the empty vector (2.09 ± 0.78 kb/ min vs. 3.05 ± 1.29 [WT], 3.18 ± 1.01 [KR]) (Fig. 2d, e). Importantly, S2-KR overexpression further increased the average replication fork progression rate by more than 1.5 fold (5.14 ± 2.3 kb per min) compared to WT and KR overexpression (Fig. 2d, e). This result suggests that cells overexpressing S2-KR may complete S-phase faster. Indeed, 10 h after PCNA overexpressing cells were released from nocodazole treatment, we observed more S2-KR-overexpressing cells transitioning into late S-phase compared to cells overexpressing WT or KR PCNA (Supplementary Fig. 3a and 3b). This result indicates that SUMO2 conjugation of PCNA positively regulates replication fork progression either by enhancing the rate of DNA synthesis or by minimizing interference that may hinder the movement of the replication fork.

**SUMO2-PCNA reduces RNAPIIo chromatin occupancy.** Given that PCNA SUMO2 conjugation is dependent on transcription, we tested the effect of SUMO2-PCNA overexpression on transcription by examining the occupancy of activated RNAPIIo on chromatin. We found that levels of CB RNAPIIo were lower in cells that overexpressed S2-KR than cells overexpressing WT and KR PCNA (Fig. 2c), even though the total RNAPII levels in the WCE of these cells were not different (Fig. 2b). The lower levels of CB RNAPIIo are consistent with lower transcription efficiency in the S2-KR-overexpressing cells, as demonstrated by reduced 5-ethynl uridine incorporation (Supplementary Fig. 3c). Because SUMO2-PCNA is only found during S-phase, we sought to determine if CB RNAPIIo was lower specifically in gene regions

that are transcribed at higher frequency during S-phase. *IMMP2L* and *WWOX* are extremely long genes for which one round of transcription requires more than one cell cycle[3]. These genes have regions transcribed at high frequency during S-phase that overlap with the FRA7K and FRA16D CFSs, respectively[3,28,29]. Therefore, we performed chromatin immunoprecipitation (ChIP) coupled with quantitative PCR (qPCR) to analyze RNAPIIo occupancy along the *IMMP2L* and *WWOX* genes in HEK293T cells overexpressing WT, KR, or S2-KR PCNA (Fig. 3a). We found that RNAPIIo levels near the promoter and 5' regions of both *IMMP2L* and *WWOX* were comparable across the three cell lines, suggesting that SUMO2-PCNA does not suppress transcription initiation (Figs 3b, c). However, in S2-KR-overexpressing cells, the association of RNAPIIo with DNA was attenuated within the internal gene regions that overlapped with the most frequent FRA7K and FRA16D breakages, especially compared to the KR-overexpressing cells (Figs 3b, c). These reduced RNAPIIo levels due to S2-KR overexpression were also observed at additional CFSs in HEK293T cells (Supplementary Fig. 4a), as well as in HeLa and HCT116 cells (Supplementary Fig. 4b, c). To further confirm that RNAPIIo is associated with CFS primarily during S-phase, we compared levels of RNAPIIo bound to FRAK7K in cells with or without nocodazole treatment. We found that in cells overexpressing either WT or the KR PCNA, the amount of RNAPIIo at FRA7K proportionally decreased as the percentage of cells in S-phase dropped after nocodazole treatment (Supplementary Fig. 4d, e). In contrast, because the RNAPIIo molecules were already destabilized, reduction in the S-phase population had little effect on RNAPIIo levels at FRA7K in S2-KR-overexpressing cells (Supplementary Fig. 4d, e). These results suggest the possibility that PCNA is conjugated with SUMO2 at the replication fork in response to nearby transcription to destabilize the binding of RNAPIIo to the chromatin. The reduced efficiency of the PCNA KR mutant to dislodge RNAPIIo may increase the frequency of RNAPIIo accumulation on chromatin, as observed in some of CFSs in the KR-overexpressing cells (Figs 3b, c; Supplementary Fig. 4a–c). The negative effect of SUMO2-PCNA on transcription may explain why it is not feasible to generate PCNA knockdown cells stably expressing a SUMO2-PCNA fusion protein.

**SUMO2-PCNA enriches CAF1 and FACT in the replisome complex.** To determine how SUMO2-PCNA attenuates RNAPII chromatin occupancy, we investigated whether SUMO2 conjugation alters protein-protein interactions. For this, we prepared CB fractions from the chromatin pellet using benzonase treatment to remove both RNA and DNA, followed by purification of the FLAG-tagged KR and S2-KR PCNA protein complexes for mass spectrometry analysis (Supplementary Fig. 5a, b). Conjugation of SUMO2 to PCNA did not significantly alter the

**Fig. 1** Transcription induces SUMO2 conjugation of PCNA at K164. **a** Schematic of the cell fractionation procedure used to separate proteins associated with transcriptionally active open chromatin (CB:RNA+) and proteins bound to not highly or non-transcribed DNA regions (CB:RNA−). **b** Western blot analysis of the indicated proteins in CB:RNA+ and CB:RNA− fractions prepared from HEK293T cells with or without 5,6-dichloro-1-beta-D-ribofuranosylbenzimidazole (DRB) treatment to inhibit transcription. The post-translationally modified form of PCNA is indicated with an asterisk (*). **c** Cell cycle analysis (by flow cytometry) of HEK293T cells after release from nocodazole (Noc) at the indicated time points. 1 C and 2 C indicate cells containing one or two copies of each chromosome, respectively. **d** Western blot analysis of PCNA and SRSF1 in CB:RNA+ and CB:RNA- fractions prepared from HEK293T cells shown in **c**. Histone H4 was used as a loading control. **e** Western blot analysis using the indicated antibodies to detect His-Myc-tagged PCNA purified under denaturing conditions from the CB:RNA+ and CB:RNA− fractions of HEK293T cells expressing His-Myc-PCNA. **f** Western blot analysis using an α-PCNA antibody to detect SUMOylated PCNA purified using Ni-NTA under denaturing conditions from whole cell extracts (WCE, center panel) and CB:RNA+ fractions (bottom panel) of HEK293T cells transfected with an empty His vector, His-SUMO1, His-SUMO2, or His-SUMO3. Unconjugated His-SUMO was detected using α-His antibody (top). Protein bands that cross-reacted with the α-PCNA antibody are indicated with a (<). **g** Western blot analysis of WT and K110R, K117R, K138R, K164R, K168R, and K254R FLAG-PCNA mutants in the CB:RNA+ and CB:RNA− fractions of HEK293T cells. Blots were probed using an α-FLAG antibody

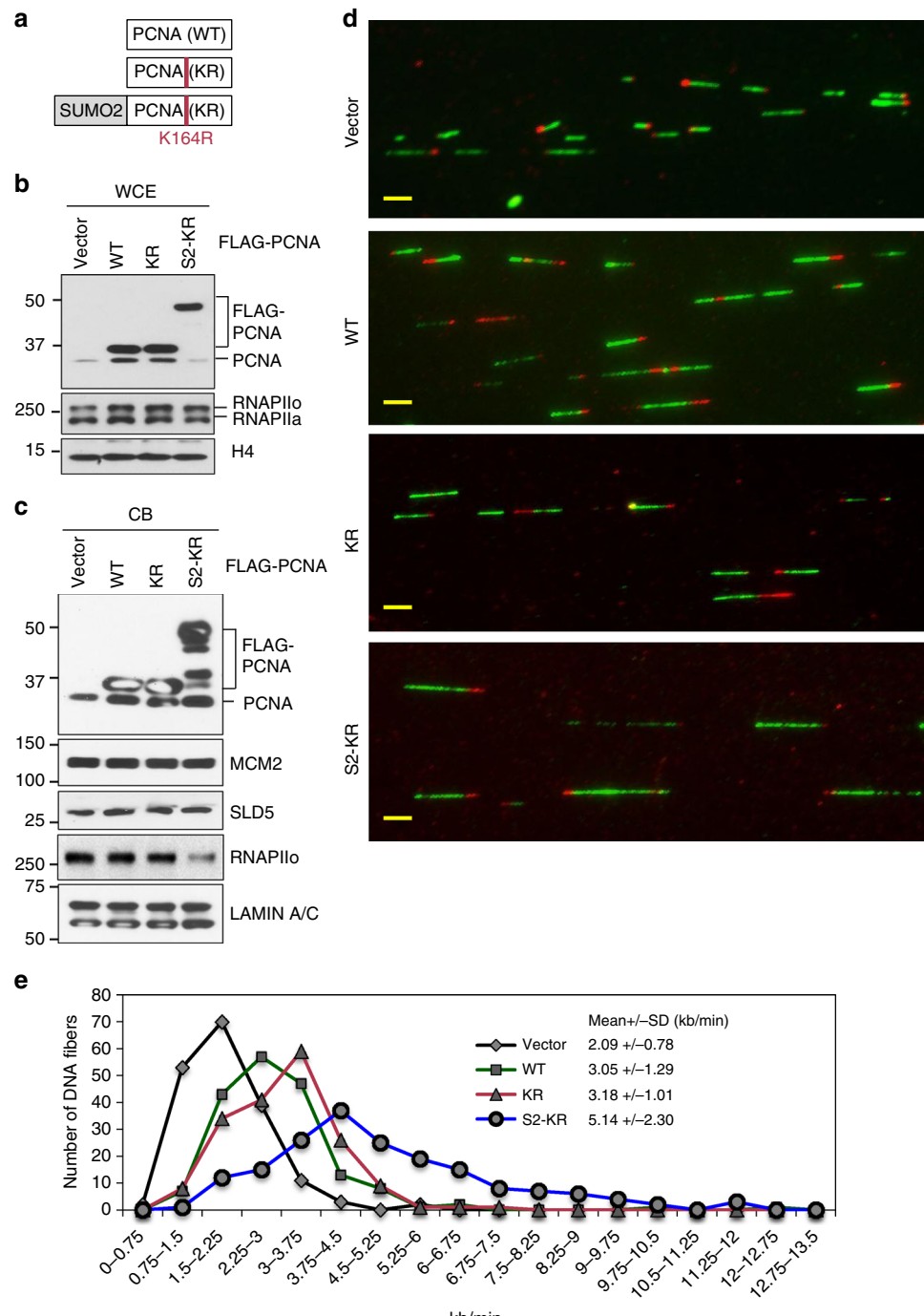

**Fig. 2** SUMO2-PCNA enhances replication fork progression. **a** Schematic diagram of FLAG-PCNA WT, K164R (KR), and SUMO2-KR (S2-KR) fusion constructs. The location of KR mutation is shown in red. **b** Western blot analysis of PCNA and FLAG-PCNA WT, KR, and S2-KR using an α-PCNA antibody and an α-RNAPII A10 antibody in whole cell extracts (WCE) prepared from HEK293T cells overexpressing the indicated FLAG proteins. Histone H4 was used as a loading control. **c** Western blot analysis of the indicated proteins in the chromatin-bound (CB) fractions of the HEK293T cells overexpressing the indicated FLAG-PCNA proteins in **b**. H5 mouse monoclonal was used to detect RNAPIIo. **d** Representative fluorescence images of DNA fibers isolated from HEK293T cells overexpressing the indicated FLAG-PCNA proteins or transfected with an empty vector. Scale bar=10 μm. **e** Distributions of the replication fork processivity rates for HEK293T cells used in **d** overexpressing the indicated FLAG-PCNA proteins or transfected with an empty vector. Fiber length was measured based on a conversion factor of 1 μm to 2.59 kb. Each average value ± standard deviation was calculated from at least 150 DNA fibers per one representative experiment. The result was reproduced in two independent assays

amounts of replication factor C (RFC), MCM2-7 helicase, FEN1, DNA polymerases, or mismatch repair factors, such as MSH6, that were co-purified with PCNA (Table 1; Supplementary Data 1). Using western blot analysis, we further confirmed that there was no significant difference between the interactions of the

KR and the S2-KR PCNA complexes with MCM2-7 or Pol δ in multiple human cell lines (Fig. 4a; Supplementary Fig. 5c, d). Instead, the largest subunit of the histone chaperone CAF1[30], CAF1A, was the most substantially and consistently enriched protein in the purified S2-KR complex identified by mass

spectrometry (Table 1; Supplementary Data 1), with considerably greater western blot band intensity for the S2-KR complex than for the KR complex (Fig. 4a; Supplementary Fig. 5c, d). In fact, mass spectrometric analysis showed that all components of CAF1, including CAF1B and RBBP4, were enriched in the S2-KR complex (Table 1, Supplementary Data 1). Using an antibody specific to CAF1A, we demonstrated that CAF1 enrichment was unique to S2-KR and not observed in the Ub-PCNA (KR) fusion protein complex (Ub-KR; Fig. 4a). To further demonstrate that SUMO2 conjugation enhanced the association of CAF1 with PCNA, we purified recombinant strep-tagged PCNA from *E. coli* (Supplementary Fig. 6a), and SUMOylated this strep-PCNA with His-SUMO2 in vitro (Supplementary Fig. 6b). Consistent with the co-immunoprecipitation experiment using the S2-KR fusion protein, using Strep-Tactin beads, substantially more CAF1 was pulled down from the human chromatin fraction by the

SUMOylated strep-PCNA than by the unmodified strep-PCNA (Fig. 4b).

In addition to CAF1, the histone chaperone FACT also functions in DNA replication via its interaction with the MCM2 subunit of the MCM2-7 replicative helicase[31–33]. This interaction facilitates the removal of parental histones ahead of the replication fork and stimulates MCM2-7 helicase to unwind the DNA template[31–34]. Although our mass spectrometric analysis did not detect an increase in FACT components in the purified S2-KR complex, western blot analysis showed that the amount of SPT16, one of two components comprising the FACT chaperone[35], was reproducibly greater in the S2-KR complex than in the PCNA WT and KR complexes (Figs 4a, b; Supplementary Fig 5c, d). We also found that limited amounts of RNAPII and the RNAPII-interacting protein RECQ5 co-purified with WT FLAG-PCNA, but these associations were enhanced in the KR

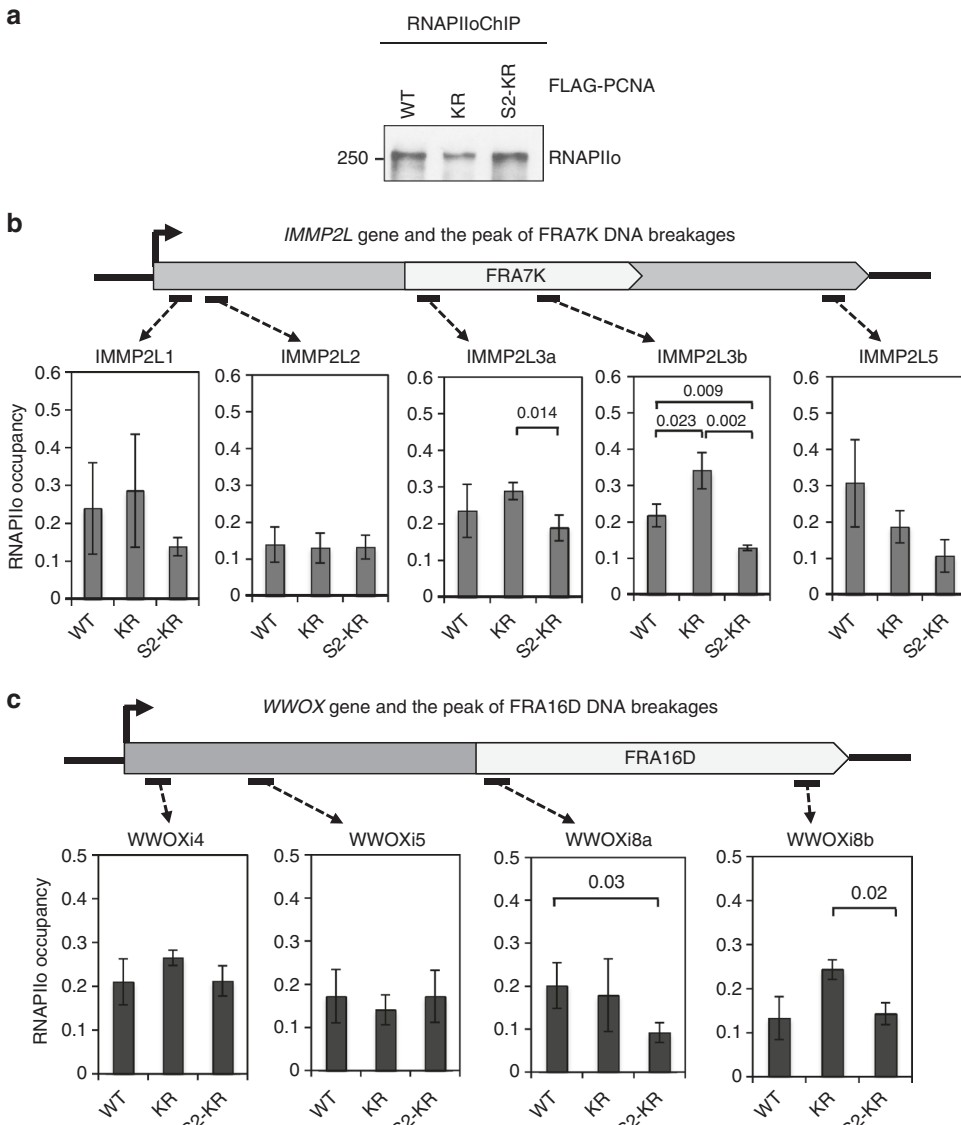

**Fig. 3** SUMO2-PCNA reduces RNAPII occupancy on chromatin. **a** Western blot analysis of immunopurified RNAPIIo using an α-RNAPII phosphor-CTD 4H8 antibody from formaldehyde-treated HEK293T cells overexpressing FLAG-PCNA WT, KR, or S2-KR fusion proteins. The blot was probed using an α-RNAPII A10 antibody. **b**, **c** (top) Schematic diagrams of regions containing DNA breaks associated with the FRA7K and FRA16D CFSs (light gray) in *IMMP2L* (**b**) and *WWOX* (**c**), respectively[28,29]. (bottom) ChIP analysis of RNAPIIo occupancy at the indicated regions of the *IMMP2L* (**b**) and *WWOX* (**c**) genes using primers derived from Helmrich et al[3] (Supplementary Table 2). Each value represents the average value ± standard deviation calculated from triplicate qPCR reactions per one representative experiment. *p* values were calculated using *t*-test analysis for statistically significant differences. Only *p* values equal to or less than 0.05 are shown. The result was reproduced in three independent assays

**Table 1 Mass spectrometry summary of the FLAG–PCNA complexes**

| Number of peptides identified by mass spectrometry | KR IP | S2-KR IP |
|---|---|---|
| PCNA (29 kDa) | 215 | 178 |
| SUMO2 (11 kDa) | 1 | 18 |
| RFC complex | | |
| RFC1 (128 kDa) | 22 | 22 |
| RFC2 (39 kDa) | 14 | 13 |
| RFC3 (41 kDa) | 3 | 9 |
| RFC4 (40 kDa) | 14 | 13 |
| RFC5 (38 kDa) | 16 | 12 |
| MCM2-7 | | |
| MCM2 (102 kDa) | 1 | 1 |
| MCM3 (91 kDa) | 5 | 4 |
| MCM4 (97 kDa) | 19 | 19 |
| MCM5 (82 kDa) | 8 | 8 |
| MCM6 (93 kDa) | 10 | 9 |
| MCM7 (81 kDa) | 19 | 22 |
| Pol δ | | |
| DPOD1 (124 kDa) | 4 | 11 |
| DPOD3 (51 kDa) | 2 | 2 |
| PDIP3 (46 kDa) | 3 | 1 |
| Pol ε | | |
| DPOE1 (261 kDa) | 4 | 5 |
| DPOE3 (60 kDa) | 1 | 0 |
| FEN1 (43 kDa) | 7 | 8 |
| LIG1 (102 kDa) | 7 | 12 |
| Mismatch repair | | |
| MSH2 (105 kDa) | 22 | 18 |
| MSH3 (127 kDa) | 7 | 7 |
| MSH6 (153 kDa) | 26 | 16 |
| CAF1 complex | | |
| CAF1A (107 kDa) | 0 | 22 |
| CAF1B (62 kDa) | 4 | 15 |
| RBBP4 (48 kDa) | 7 | 9 |

Numbers of peptides detected by mass spectrometry for each of the indicated proteins co-purified with KR and the S2-KR FLAG–PCNA protein complexes isolated from the chromatin-bound (CB) fractions of HEK293T cells expressing the indicated FLAG-tagged proteins

complex (Fig. 4a). The greater association of RNAPII and RECQ5 with the KR mutant may reflect a higher frequency of physical encounters between the replisome and the RNAPII complex. In contrast, these associations were nearly absent for the S2-KR complex, likely because the association of RNAPII with chromatin was destabilized.

**CAF1A and SSRP1 interact with SUMO2-PCNA via their SIMs**. CAF1A contains two PIP motifs, but only the internal PIP2 is required for CAF1-mediated nucleosome assembly[36]. However, PIP2 alone exhibits weak affinity to PCNA[36]. Because CAF1A also contains a SIM at its N-terminus[37], we wondered if the CAF1A SIM is required to interact with the SUMO2 moiety of PCNA for enhancing the CAF1-PCNA interaction. To test this, we expressed FLAG-tagged WT and SIM mutant CAF1A in HEK293T cells and purified from the CB fraction on M2 agarose beads. When we incubated these beads with recombinant strep-PCNA or strep-SUMO2-PCNA (S2-PCNA) fusion proteins purified from *E. coli*, we found that more S2-PCNA than unmodified PCNA molecules were pulled down by FLAG-CAF1A WT. However, SIM mutation greatly reduced CAF1A interactions with S2-PCNA with little effect on unmodified PCNA (Fig. 4c). Similar pull-downs using FLAG-CAF1A purified from *E. coli* showed the same result (Fig. 4d), confirming that the CAF1A SIM is involved in stabilizing interactions of CAF1 with SUMO2-PCNA. These results are consistent with the previous

observation that CAF1A SIM preferentially interacts and localizes with SUMO2/3 at replication forks[37].

We also tested whether a SIM is involved in enhancing the interaction between FACT and SUMO2-PCNA. Because the SSRP1 subunit of FACT contains both a PIP variant ([82]FEKLSDFF[89]) and a SIM ([273]LILLF[277]), we expressed WT or SIM mutant FLAG-SSRP1 in HEK293T cells and isolated their CB fractions. Similar to endogenous SPT16 component of FACT, WT FLAG-SSRP1 was preferentially pulled down by strep-S2-PCNA compared to PCNA on the Strep-Tactin beads, but this interaction was abolished by the SSRP1 SIM mutation (Fig. 4e). In summary, these results indicate the SUMO2-PCNA enriches CAF1 and FACT in the replication complex via its interactions with CAF1A and SSRP1 SIMs, respectively.

**SUMO2-PCNA promotes repressive chromatin**. PCNA is known to promote CAF1-dependent histone deposition activity[36,38]. Therefore, to determine if the greater association of SUMO2-PCNA with CAF1 enhances histone deposition, we used a DNA supercoiling assay to analyze the histone deposition activity of cell extracts prepared from HEK293T cells overexpressing WT, KR, or S2-KR PCNA. Using histone proteins purified from HeLa nuclei by acid extraction[39], we detected more DNA supercoiling induction by histone deposition in the reaction containing HEK293T cell extracts overexpressing S2-KR compared to reactions containing extracts from cells overexpressing WT and KR (Fig. 5a, left panel). However, this enhancement was not observed when the reactions were performed using pre-assembled H3-H4 tetramers (Fig. 5a, right panel), suggesting that the greater histone deposition activity in S2-KR-overexpressing cells is driven by CAF1, which specifically binds to histone H3-H4 dimers to promote nucleosome assembly[40]. Indeed, immunodepletion of CAF1 (Supplementary Fig. 6c) abolished the enhanced histone deposition activity observed in the S2-KR-overexpressing cell extracts (Fig. 5b), providing further evidence that SUMO2-PCNA enhances CAF1 histone deposition activity.

CAF1 specifically deposits histone H3.1 variant[41,42]. Therefore, we analyzed the effect of SUMO2-PCNA on the amount of histone H3.1 variant in the chromatin. To do this, we expressed either HA-tagged histone H3.1 or H3.3, the latter of which has been implicated in open chromatin maintenance to aid transcription and is not a CAF1 substrate[43–45], in cells overexpressing the various PCNA constructs and performed ChIP analysis using an HA-tag antibody to measure the amounts of H3.1 and H3.3 at CFSs (Fig. 5c). Consistent with the enhanced CAF1-dependent histone deposition activity by SUMO2-PCNA (Fig. 5b), we found that H3.1 occupancy was increased at multiple CFSs in S2-KR-overexpressing cells as compared to WT- and KR-overexpressing cells (Figs 5d, e; Supplementary Fig. 7). Elevated H3.1 levels were accompanied with diminished H3.3 occupancy in chromatin at some examined loci (Figs 5d, e; Supplementary Fig. 7). In addition, CAF1-mediated H3.1 deposition also facilitates H3K9me2/3 and H3K27me3 repressive histone marks[41,42]. Indeed, we found that overexpressing either CAF1 or S2-KR in cells increased the levels of H3K9me3 histone marks as compared to overexpressing WT or KR PCNA (Figs 5f, g), strongly supporting a synergistic role of CAF1 activity with SUMO2-PCNA in antagonizing transcription. Consistent with a more repressive state, chromatin from S2-KR-overexpressing cells was more resistant to micrococcal nuclease (MNase) digestion (Fig. 5h), a method for analyzing chromatin accessibility[46,47]. These results indicate that SUMO2-PCNA facilitates CAF1-dependent deposition of repressive histones and reduces open chromatin structure, a phenomenon that is consistent with the destabilization of RNAPIIo from chromatin.

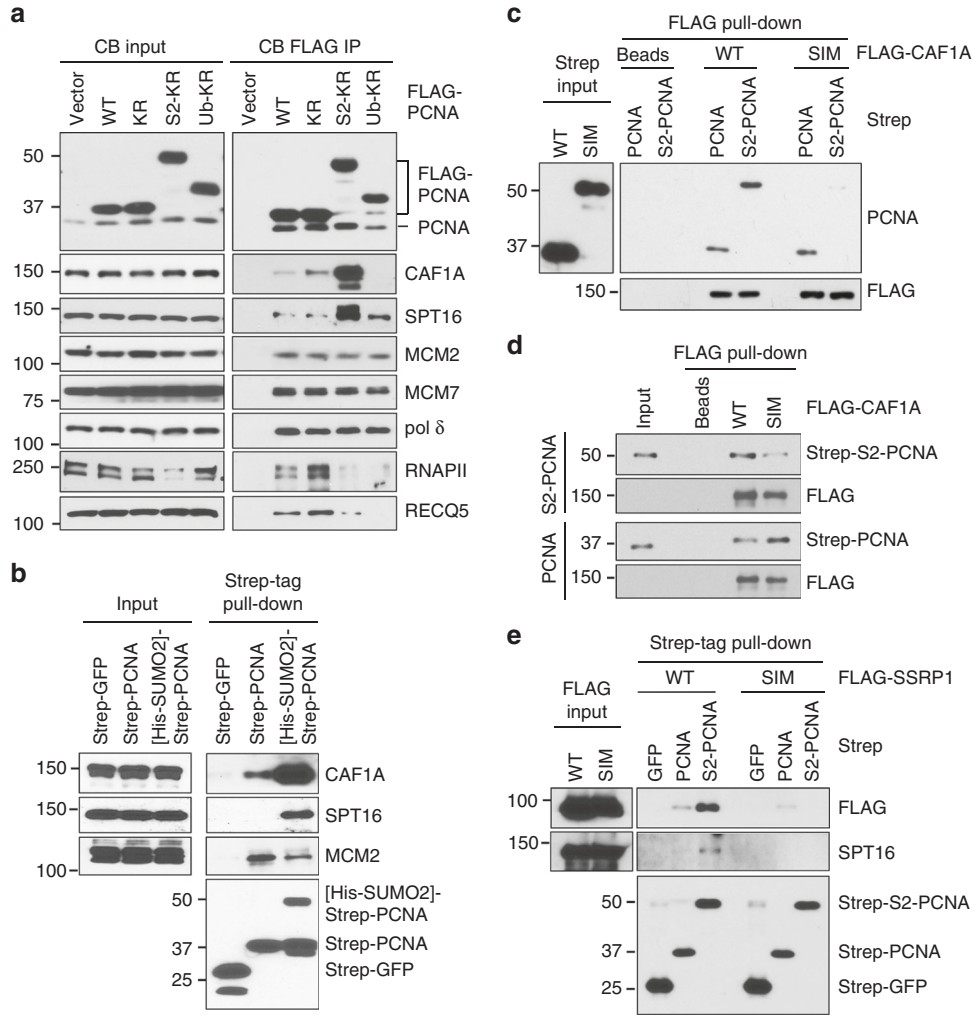

**Fig. 4** SUMO2-PCNA enriches CAF1 and FACT within the replication complex. **a** Western blot analysis of the indicated proteins in the CB fractions (left) and the FLAG-PCNA complexes purified from CB fractions (right) prepared from cells expressing the indicated FLAG-tagged proteins. Ub-KR indicates the FLAG-ubiquitin-PCNA (KR) fusion protein. **b** Western blot analysis of the indicated proteins in the CB input (left panels) used for the in vitro pull-down experiments of purified recombinant strep-tagged PCNA, SUMO2-conjugated PCNA, and GFP anchored to Strep-Tactin resins (right panels). SUMO2-modified PCNA was generated by the in vitro SUMOylation reaction using strep-PCNA and His-SUMO2 shown in Supplementary Fig. 6a, b. **c** Western blot analysis of the input levels of recombinant strep-PCNA and strep-SUMO2-PCNA fusion (S2-PCNA) purified from *E. coli* (left panel) and of strep-PCNA and strep-S2-PCNA pulled down by WT or SIM mutant FLAG-CAF1A proteins purified and anchored on FLAG M2 agarose beads from HEK293T cells expressing the corresponding constructs (right panels). **d** Western blot analysis of strep-S2-PCNA (upper 2 panels) and strep-PCNA (lower 2 panels) pulled down by FLAG-CAF1A purified from *E. coli* and anchored on FLAG M2 agarose beads. **e** Western blot analysis of the indicated proteins in the CB input prepared from HEK293T cells expressing FLAG-SSRP1 WT or SIM mutant (left panels) and of FLAG-SSRP1 and SPT16 pulled down from the CB fractions in vitro using purified recombinant strep-tagged PCNA, S2-PCNA, and GFP anchored to Strep-Tactin resins (right panels)

**SUMO2-PCNA suppresses transcription-induced DSBs**. Next, we wondered if the reduction in RNAPIIo chromatin association by the SUMO2-PCNA-containing replication complex functions to minimize TRC-induced DSBs. Indeed, we found that overexpression of the KR PCNA mutant led to a greater number of γH2AX foci, which indicate the presence of DNA damage, than the overexpression of WT PCNA (Fig. 6a–c; Supplementary Fig. 8a, b). Neutral comet assays confirmed that KR-overexpressing cells accumulated DSBs, as demonstrated by the increased average as well as the maximum of tail moments (Fig. 6d). Treatment with DRB partially suppressed KR overexpression-induced DSB formation (Fig. 6b–d; Supplementary Fig. 8), suggesting that a substantial number of DSBs formed in the KR-overexpressing cells are caused by RNAPII-dependent transcription. Importantly, overexpression of the fusion of SUMO2 to the KR mutant led to fewer DSBs compared to overexpression of the unmodified KR (Fig. 6a–d; Supplementary Fig. 8). DRB had little effect on the residual DSBs formed in S2-KR-overexpressing cells (Fig. 6b–d; Supplementary Fig. 8), suggesting that conjugation of SUMO2 to PCNA specifically suppresses the formation of DSBs induced by RNAPII-dependent transcription. The remaining DSBs found in the S2-KR-overexpressing cells likely result from defects in other PCNA K164 modifications. Using ChIP analysis (Fig. 6e, f), we further showed that γH2AX molecules accumulated at the FRA7K CFS (IMMP2L3b) in cells that overexpressed the KR mutant. This accumulation was suppressed in S2-KR-overexpressing cells (Fig. 6f). In the KR-overexpressing cells, transcription-dependent γH2AX molecules accumulated at FRA7K at a higher frequency than at IMMP2L5, a region of the *IMMP2L* gene located outside of the CFS (Fig. 6g). And whereas the number of γH2AX molecules found at IMMP2L3b

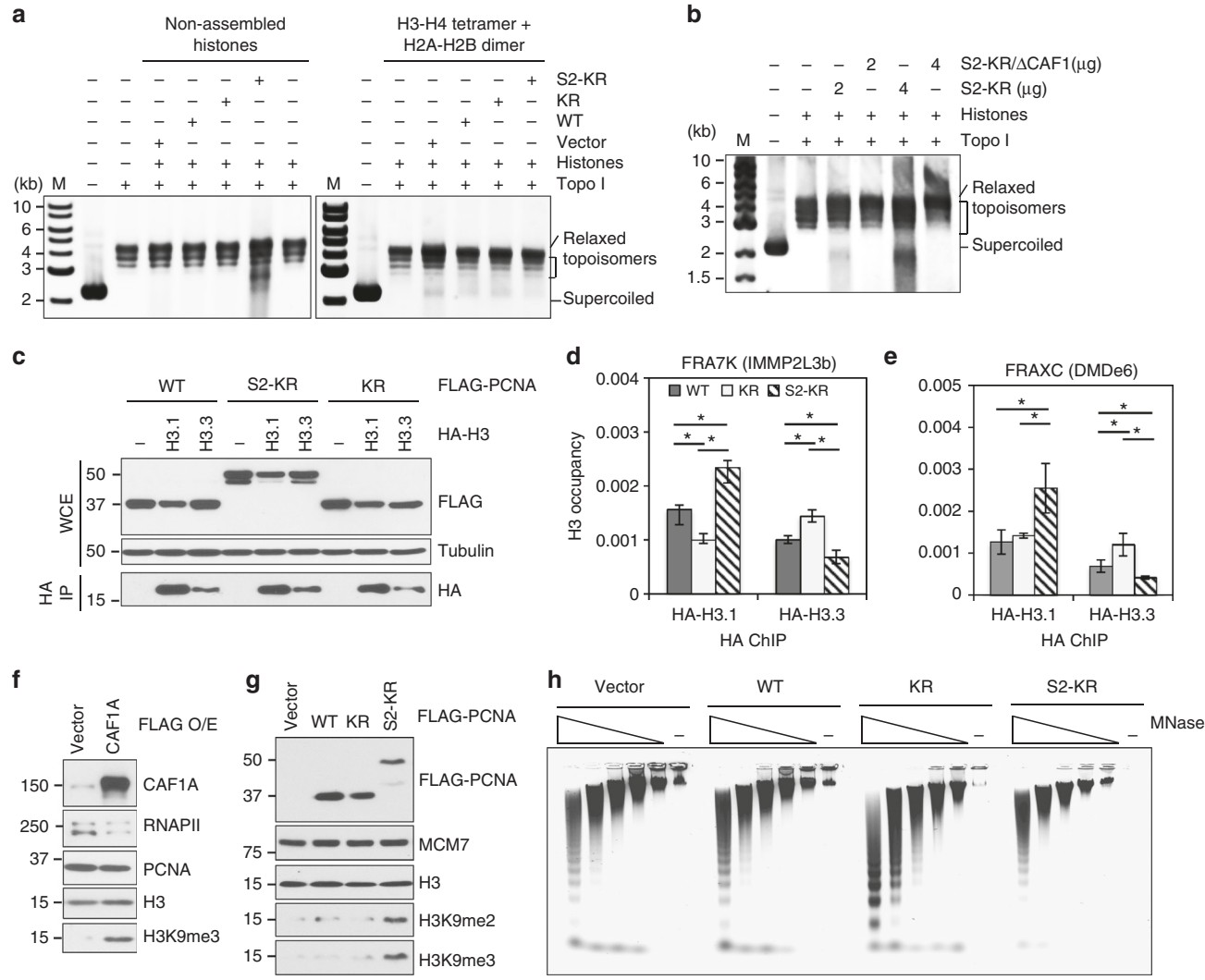

**Fig. 5** SUMO2-PCNA enhances deposition of CAF1-dependent histone H3.1. **a** DNA supercoiling assays to detect nucleosome assembly. Reactions were initiated by mixing cytoplasmic extracts (6 µg each) from HEK293T cells transfected with a control vector, or WT, K164R (KR), or SUMO2-KR (S2-KR) PCNA with purified, non-assembled histones (left panel) or pre-assembled H3-H4 tetramer and H2A-H2B dimer (right panel) in a reaction containing relaxed circular DNA plasmids (200 ng, linear length: 3.0 kb) pre-treated with topoisomerase I (Topo I). After deproteination to stop the reaction, DNA molecules were precipitated and analyzed by agarose gel electrophoresis. Supercoiled DNA, relaxed DNA, and topoisomers are indicated. **b** DNA supercoiling assays to detect nucleosome assembly using the indicated amounts of control or CAF1-depleted cytoplasmic extracts from S2-KR-overexpressing HEK293T cells. **c** (top) Western blot analysis of expression of the indicated FLAG-PCNA complexes in whole cell extracts (WCE) of HEK293T cells co-transfected with the indicated HA-tagged histone H3 or vector alone. Tubulin was used as a loading control. (bottom) Western blot analysis of HA-H3 immunopurified from the indicated cells with an HA antibody. **d**, **e** ChIP analysis of the samples from (**c**) for HA-H3.1 and HA-3.3 occupancy at the FRA7K (**d**) and FRAXC (**e**) gene regions. Each average value ± standard deviation was calculated from triplicate qPCR reactions per one representative experiment. *p* values were calculated by t-test analysis for statistically significant differences. *p* values equal to or less than 0.05 are indicated with an asterisk (*).The result was reproduced in three independent assays. **f** Western blot analysis of CAF1, RNAPII, histone H3, and H3 tri-methyl lysine 9 (H3K9me3) in WCE of HEK293T cells with or without FLAG-CAF1 overexpression. **g** Western blot analysis of FLAG-PCNA, MCM7, histone H3, and H3 di-methyl K9 (H3K9me2) and H3K9me3 in WCE of HEK293T cells transfected with the indicated FLAG-PCNA constructs. **h** Micrococcal nuclease (MNase) digestion assay of nuclei isolated from HEK293T cells overexpressing the indicated FLAG–PCNA proteins. The digested samples were deproteinized, and the DNA purified and analyzed by agarose gel electrophoresis

was lower for cells treated with DRB, the number of γH2AX molecules on the non-transcribed region upstream of the *ACTB* gene (ACTBUPS) was not affected by DRB treatment of the KR-overexpressing cells (Fig. 6g). In addition to RNAPII molecules, it has been shown that unresolved replisome-RNAPII collisions can lead to the formation of R-loops[48,49], which can also block the replication fork[50]. However, when we carried out DNA:RNA immunoprecipitation (DRIP) in KR overexpressing cells using an R-loop specific antibody, S9.6, we did not observe a significant R-loop accumulation at CFSs

FRA7K (Fig. 6h, left), FRA16D (not shown) or exon 3 of the *ACTB* gene (Fig. 6h, right), the latter of which is prone to R-loop formation[21]. Most likely, in the KR overexpressing cells, R-loops generated from the unresolved TRCs were efficiently removed by topoisomerases, RNase H1 and DNA helicases, such as BLM[21,51–55], all of which are expected to be functional in these cells. Collectively, these data indicate that the accumulation of RNAPII at the collision site is the primary cause for the TRC-induced genomic instability in cells defective in PCNA SUMO2 conjugation.

**SUMO2 conjugation of PCNA is dependent on RECQ5**. We next determined how RNAPII induces SUMO2 conjugation of PCNA at the replication fork. The RNAPII-interacting protein RECQ5, which is known to directly interact with PCNA[18], behaves similarly to RNAPII in its interactions with the various forms of PCNA on human chromatin (Fig. 4a). Therefore, we wondered if RECQ5 is involved in the induction of PCNA SUMO2 conjugation. Indeed, we found that the level of SUMO2-PCNA in the CB:RNA+ fraction was lower in the RECQ5 siRNA knockdown cells than in scrambled-siRNA-treated cells (Fig. 7a, b). As expected, treatment with the DNA damaging agent camptothecin (CPT) had no effect on the induction of SUMO2-PCNA in control or RECQ5 knockdown cells (Fig. 7b). These results

indicate that the level of SUMO2-PCNA is dependent on RECQ5 but independent of DNA damage. To further support a role for RECQ5 in promoting SUMO2-PCNA, we overexpressed FLAG-tagged RECQ5 and found that the level of SUMO2-PCNA on the chromatin was greatly enhanced in RECQ5-overexpressing cells (Fig. 7c). Although endogenous RECQ5 primarily associates with transcribed chromatin[21], the exogenously expressed RECQ5 molecules were detected in both the CB:RNA+ and CB:RNA− fractions due to their over-abundance in RECQ5-overexpressing cells (Fig. 7c, d, upper panel). As a consequence, SUMO2-PCNA induction was no longer restricted to the CB:RNA+ fraction, and was also found, though to a lesser extent, in the CB:RNA- fraction (Fig. 7c, d, top panels).

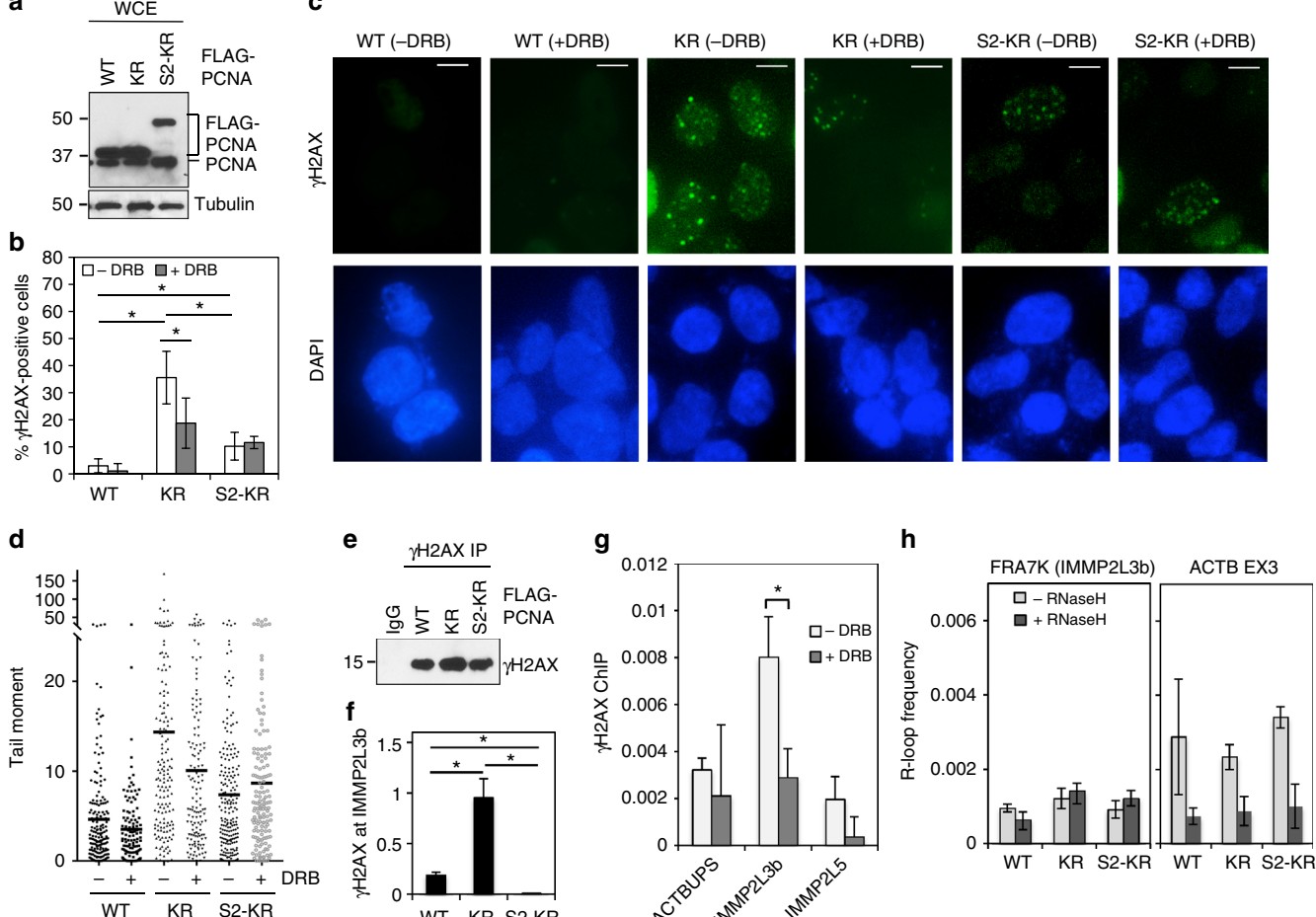

**Fig. 6** SUMO2-PCNA suppresses transcription-induced DSBs. **a** Western blot analysis of PCNA and FLAG-PCNA in whole cell extracts (WCE) using an antibody against PCNA. Tubulin was used as a loading control. **b** Quantification of γH2AX-positive cells overexpressing the indicated FLAG-PCNA complexes with or without DRB treatment (100 μM). Each value in the graph represents the average value ± standard deviation (n > 200) per one representative experiment. Only cells with 5 or more γH2AX foci were counted as positive cells. The result was reproduced in two independent assays. **c** Representative fluorescence images of γH2AX foci (green) in HEK293T cells overexpressing the indicated FLAG-PCNA proteins with or without DRB treatment. Nuclei were counterstained with DAPI. Scale bar, 10 μm. **d** The average tail moments (horizontal lines) measured by the neutral comet assay using HEK293T cells overexpressing the indicated FLAG-PCNA constructs with or without DRB treatment. Each dot represents the tail moment of a single cell. At least 150 cells were analyzed per one representative experiment. The result was reproduced in three independent comet assays. **e** Western blot analysis of immunopurified γH2AX from formaldehyde-treated HEK293T cells for ChIP analysis. **f** ChIP analysis of the samples from **e** for γH2AX occupancy at FRA7K using IMMP2L3b primers. The result was reproduced in two independent assays. **g** ChIP analysis of γH2AX occupancy at indicated loci in HEK293T cells overexpressing FLAG-PCNA (KR) and treated with or without DRB. The result was reproduced in three independent assays. **h** Representative analysis of R-loop frequency at IMMP2L3b (left) and ACTB exon 3 (EX3; right) sites in HEK293T cells overexpressing the indicated FLAG-PCNA constructs. The isolated nucleic acids were treated with or without RNase H prior to R-loop immunoprecipitation. The result was reproduced in two independent assays. For all the ChIP experiments described above, each value represents the average value±standard deviation calculated from triplicate qPCR reactions per one representative experiment. p values were calculated using t-test analysis for statistically significant differences. p≤0.05 are indicated with an asterisk (*)

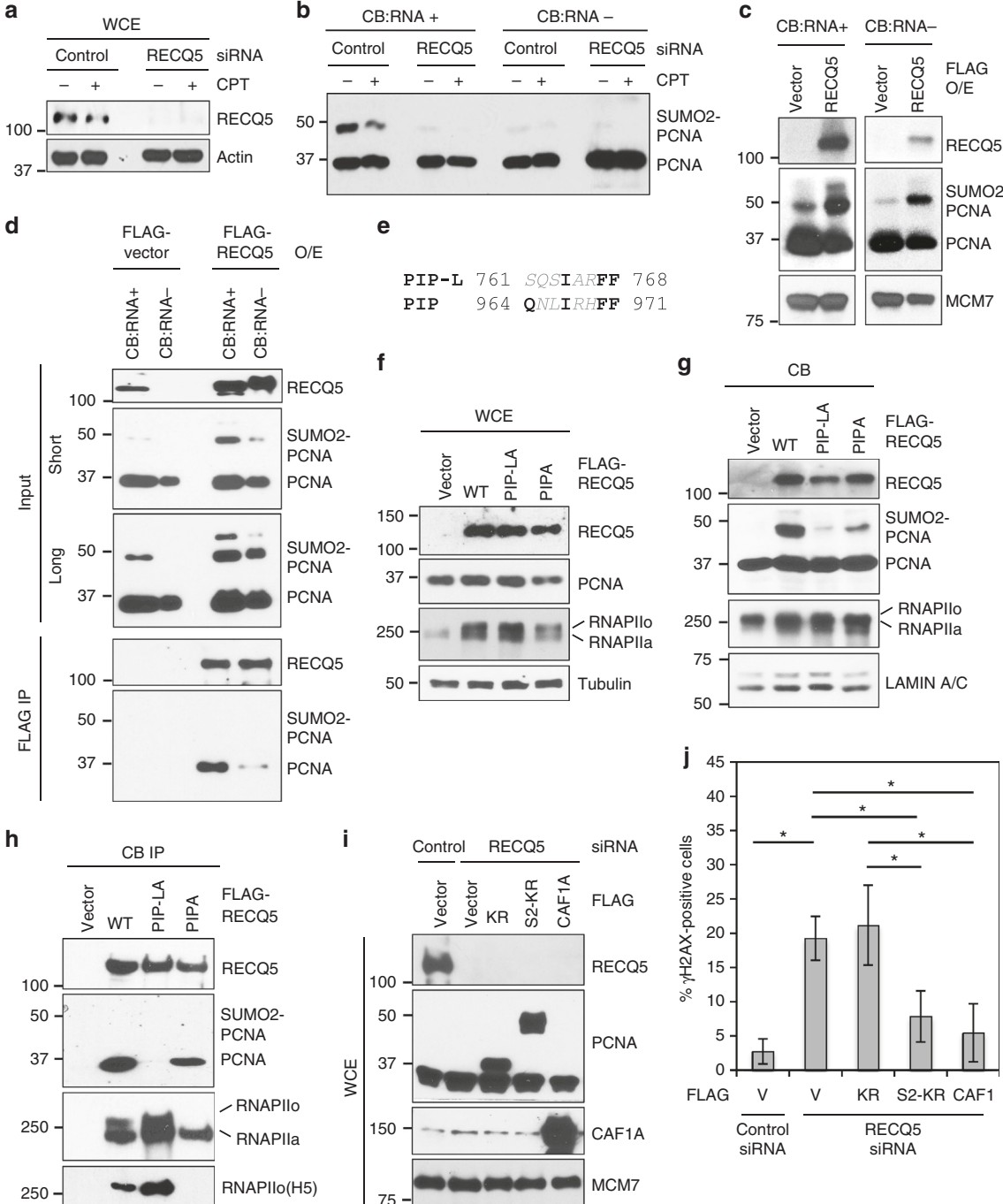

**Fig. 7** RECQ5 promotes SUMO2 conjugation of PCNA. **a** Western blot analysis of RECQ5 in whole cell extracts (WCE) prepared from control or RECQ5 siRNA knockdown HEK293T cells treated with or without camptothecin (CPT, 5 μM). Actin was used as a loading control. **b** Western blot analysis of PCNA in the CB:RNA+ and CB:RNA− fractions prepared from HEK293T cells with or without CPT treatment from **a**. **c** Western blot analysis of the indicated proteins in CB:RNA+ and CB:RNA− fractions prepared from HEK293T cells with or without exogenous overexpression of FLAG-RECQ5. MCM7 was used as a loading control. **d** Western blot analysis of the indicated proteins in the input of the CB:RNA+ and CB:RNA− fractions with or without exogenous overexpression of FLAG-RECQ5 visualized with short and long exposure times (top) and in the FLAG-RECQ5 complex purified from the CB:RNA+ and CB:RNA− fractions shown above (bottom). **e** Amino acid sequences of the PCNA-Interacting-Protein (PIP) and PIP-like (PIP-L) motifs found in human RECQ5 protein are shown in bold. **f** Western blot analysis of the indicated proteins in WCE prepared from HEK293T cells with or without exogenous overexpression of FLAG-RECQ5 WT and mutant proteins. **g** Western blot analysis of the indicated proteins in the chromatin-bound (CB) fractions prepared from HEK293T cells from **f**. **h** Western blot analysis of the indicated proteins in the FLAG-RECQ5 complexes purified from CB fractions shown in **g**. **i** Western blot analysis of the indicated proteins in WCE prepared from control or RECQ5 siRNA knockdown HEK293T cells with exogenous overexpression of indicated FLAG-tagged PCNA constructs or CAF1 or treated with an empty vector (V). **j** Quantification of γH2AX-positive cells indicated in **i**. Each value in the graph represents the average value ± standard deviation ($n > 200$) per one representative experiment. Only cells with 5 or more γH2AX foci were counted as positive cells. $p$ values equal to or less than 0.05 are indicated with an asterisk (*).The result was reproduced in two independent assays

We further observed that RECQ5 interacted only with non-modified PCNA, and the amounts of PCNA co-purified with FLAG-RECQ5 in the CB:RNA+ and CB:RNA− fractions (Fig. 7d, bottom panel) correlated with SUMO2-PCNA levels (Fig. 7d, 2nd panel), suggesting that the interaction of RECQ5 with non-modified PCNA may be involved in promoting PCNA SUMO2 conjugation. A canonical PIP motif located between residues 964–971 of RECQ5 (Fig. 7e) has been reported[18]. Indeed, we found that mutation of the PIP sequence (PIPA: Q964A, I967A, F970A, and F971A) resulted in lower SUMO2-PCNA induction by RECQ5 overexpression compared to the WT RECQ5 (Fig. 7f, g). However, surprisingly, the PIPA mutation abolished the interaction of RECQ5 with RNAPIIo but not with PCNA or RNAPIIa (Fig. 7h). The absence of RNAPIIo in the RECQ5 PIPA mutant complex was further confirmed by western blot analysis using a phosphor-RNAPII CTD antibody (H5; Fig. 7h). The effect of the PIPA mutation on the RECQ5-RNAPIIo interaction is most likely due to the fact that the PIP motif overlaps with the Set-RPB1-interaction (SRI) domain, which is located between residues 922–991 of RECQ5 and is important for this interaction[17,56]. These results not only indicate that the RECQ5–RNAPIIo interaction is critical for PCNA SUMO2 conjugation, but also argue that this canonical PIP is not the primary PCNA-interaction motif of RECQ5.

Our further sequence analysis identified a PIP-like (PIP-L) motif between residues 761–768 of RECQ5 (Fig. 7e), lacking the conserved glutamate (Q) of PIP, similar to the PIP-L motifs found in the translesion polymerases η and ι[57]. We introduced mutations to the RECQ5 PIP-L motif (PIP-LA: I764A, F767A, and F768A) that alone were sufficient to abolish the RECQ5-PCNA interaction

(Fig. 7h). Importantly, overexpressing PIP-LA mutant FLAG-RECQ5 failed to enhance PCNA SUMO2 conjugation (Fig. 7g). Interestingly, we also observed that interactions with RNAPIIo were stronger for the PIP-LA mutant RECQ5 than for the WT RECQ5 protein (Fig. 7h). It is possible that RNAPIIo bound by the RECQ5-PIP-LA mutant is not dislodged from the chromatin, resulting in RNAPIIo–RECQ5 complex accumulation on the DNA.

Previously, we showed that RECQ5 knockdown cells accumulate transcription-mediated DSBs during S-phase[17]. Given the necessity of RECQ5 to induce PCNA SUMO2 conjugation, we next wondered if a defect in this modification contributes to the formation of transcription-mediated DSBs in RECQ5-deficient cells. Indeed, we found that RECQ5 knockdown cells exhibited increased levels of spontaneous γH2AX foci, and these γH2AX foci were partially suppressed by the overexpression of S2-KR or CAF1A, but not of the KR mutant or Ub-KR fusion proteins (Fig. 7i, j; Supplementary Fig. 9). These results identify RECQ5 as a key mediator that physically links RNAPIIo to PCNA. They also suggest that the simultaneous interactions of RECQ5 with RNAPIIo via the SRI domain and with PCNA through the PIP-L motif are required for PCNA SUMO2 conjugation and the suppression of TRC-induced DSBs through CAF1-dependent chromatin remodeling.

## Discussion

Through this study, we discovered the specific association between K164 SUMO2-conjugated human PCNA and transcribed chromatin (Fig. 1). PCNA SUMO2 conjugation, which occurs during DNA synthesis but is induced by transcription (Fig. 1), is important for minimizing TRC-induced DSBs

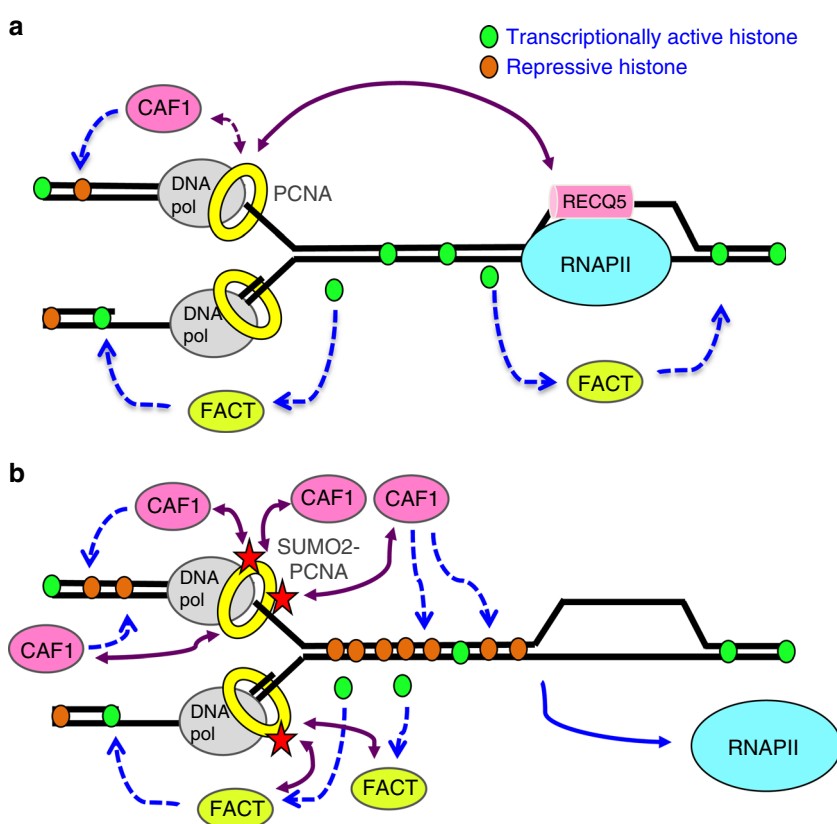

**Fig. 8** Proposed model for SUMO2-PCNA-mediated TRC resolution. **a** Simultaneous interactions of RECQ5 with RNAPIIo and PCNA induce PCNA SUMO2 conjugation. Black lines represent DNA. Purple double arrows indicate protein–protein interactions. Single dashed arrows present histone chaperone activity. **b** SUMOylated PCNA enriches CAF1 and FACT in the replication complex to deposit repressive histones and reduce chromatin accessibility. Stars represent SUMO2. See text for details

associated with CFS instability (Fig. 6). Consistent with the role of SUMO2-PCNA in resolving TRCs, we demonstrated that the SUMO2 conjugation of PCNA enhances replication fork progression while simultaneously reducing the chromatin association of RNAPIIo (Figs. 2 and 3). We further found that a key function of SUMO2-PCNA is to regulate nucleosome composition by enhancing repressive histone marks (Fig. 5), and this is achieved by enriching CAF1 and FACT histone chaperones in the replisome complex (Fig. 4).

Based on our observations, we propose the following model for SUMO2-PCNA displacement of RNAPII to resolve TRC (Fig. 8): the increased interactions between CAF1 and SUMO2-PCNA enhance CAF1-dependent histone deposition activity, thereby contributing to increased repressive histone marks and reduced chromatin accessibility (Fig. 5), which is consistent with the destabilization of RNAPII from the chromatin. Hence, cells that are prone to TRC-induced DSBs due to defective PCNA SUMO2 conjugation, such as RECQ5 knockdown cells (Fig. 7), can be rescued by overexpressing either SUMO2-PCNA or CAF1. While our study did not distinguish between a head-on and a co-directional collisions, we suggest that this mechanism of displacement can be applied to both types of collisions. Alternatively, because the forward movement of a replisome is capable of displacing RNAPII from DNA in vitro[1,58], SUMO2-PCNA might also reduce the chromatin association of RNAPII by enhancing the speed of replication fork progression via the enrichment of FACT, which removes parental histones ahead of the replication fork and stimulates MCM2-7 helicase activity to unwind DNA[33,59]. To support and sustain this faster replication fork progression, enhanced histone removal ahead of the replication fork must be accompanied by increased histone deposition to prevent the uncoupling of DNA synthesis from nucleosome assembly behind the replication fork[33]. Therefore, the SUMO2-PCNA-mediated enrichment of CAF1 in the replication complex and elevated CAF1-dependent histone deposition activity would be expected to also contribute to enhanced replication fork progression.

We showed that PCNA SUMO2 conjugation is induced by mechanisms distinct from those inducing SUMO1 conjugation. Therefore, it is likely that distinct SUMO E3 ligases with different specificities toward various SUMO isoforms and different responses to various cellular stimuli are required for conjugating SUMO1 and SUMO2 to PCNA. However, since the first observation of PCNA SUMO1 conjugation in human cells in 2012 and subsequently SUMO2 conjugation in 2015[11–13], the identities of these SUMO E3 ligases remain unknown, largely due to the fact that PCNA can be efficiently SUMOylated in vitro without an E3 ligase[60,61]. In our study, we discovered that RECQ5 is a positive regulator of PCNA SUMO2 conjugation (Fig. 7). Previously, we showed that RECQ5, together with RNA splicing factor SRSF1, functions as a co-factor for the SUMO E3 ligase PIAS1 to facilitate SUMOylation of topoisomerase I during transcription[21]. It is possible that in vivo, the cellular environment necessitates the involvement of this PIAS1-SRSF1-RECQ5 E3 ligase complex to promote PCNA SUMOylation. RECQ5, which we have shown primarily associates with RNAPII at transcribed chromatin, may serve as a crucial sensor for the replication fork to detect nearby transcription machinery. The proximity of RNAPII to the replication fork allows PCNA to interact with the RNAPII-associated RECQ5 (Fig. 8), which recruits PIAS1, or a yet-to-be-identified SUMO E3 ligase, to trigger SUMO2 conjugation of PCNA. Clearly, identifying the SUMO E3 ligase that functions synergistically with RECQ5 to promote PCNA SUMO2 conjugation will allow us to better understand how SUMO2 conjugation of

PCNA is induced by transcription via pathways distinct from those that signal SUMO1 conjugation.

## Methods

**Plasmids**. pET11-SUMO2 was kindly provided by Dr. Yuan Chen (City of Hope) and was used for bacterial His-SUMO2 expression and purification. pcDNA3-6His plasmids containing SUMO1, SUMO2, and SUMO3 were generous gifts from Dr. Ronald Hay (University of Dundee). The PCNA cDNA (with stop codon) and Ub cDNA (without stop codon) were PCR-amplified from a HeLa cDNA library. The PCNA cDNA was then cloned into pBiFC-VN173 (Addgene) between the SalI and XbaI sites to generate an N-terminal FLAG-tagged PCNA mammalian expression construct. FLAG-PCNA K110R, K117R, K138R, K164R, K168R, and K254R mutants were created by site-directed mutagenesis using pBiFC-VN173-PCNA as the template. The primer sequences used for the mutagenesis are shown in Supplementary Table 1. To generate the S2-KR fusion construct, SUMO2 cDNA was PCR-amplified from pET11-SUMO2 and inserted into pBiFC-VN173-PCNA between the HindIII and BglII sites. To generate the Ub-PCNA fusion construct, the Ub cDNA was inserted into pBiFC-VN173-PCNA between the HindIII and EcoRI sites. For expression of mammalian His-myc-tagged PCNA, the PCNA cDNA was cloned into the pcDNA3.1 myc-His vector between the BamHI and XhoI sites. Mammalian HA-H3.1 and HA-H3.3 expression constructs were generated by inserting the H3.1 or H3.3 cDNA into the pCMV6-Entry vector (OriGene) between the BamHI and MluI sites. The C-terminal FLAG sequence was replaced with HA using the following annealed oligonucleotides: 5′-GGC CGC TCT ACC CAT ACG ATG TTC CAG ATT ACG CTG CTG CTT ACC CAT ACG ATG TTC CAG ATT ACG CTG TTT-3′ and 5′-AAA CAG CGT AAT CTG GAA CAT CGT ATG GGT AAG CAG CAG CGT AAT CTG GAA CAT CGT ATG GGT AGA GC-3′. To generate the Strep-PCNA, Strep-S2-PCNA fusion, and Strep-GFP constructs for bacterial expression and purification, the His-tag sequence of pET16b was replaced with 2 × StrepII tag to generate the pET-Strep plasmid using the following annealed oligonucleotides: 5′-CAT GGA TGT GGA GCC ACC CGC AGT TCG AAA AAA GCT GGA GCC ACC CGC AGT TCG AAA AAG CAC A-3′ and 5′-TAT GTG CTT TTT CGA ACT GCG GGT GGC TCC AGC TTT TTT CGA ACT GCG GGT GGC TCC ACA TC-3′. PCNA or GFP cDNA was then cloned into the pET-Strep plasmid between the BamHI and XhoI sites. The DNA fragment containing SUMO2-PCNA fusion was cloned into pET-Strep plasmid between NdeI and BamHI sites. Human SSRP1 cDNA from Dr. Xiaochun Yu (City of Hope) was PCR-amplified and cloned into a pCMV-FLAG vector with NdeI and XhoI sites. Human CAF1A cDNA was PCR-amplified from a HeLa cDNA library and cloned into pET16b-FLAG[62] and pCMV-FLAG vectors with NdeI and XhoI sites. All plasmid sequences were confirmed by DNA sequencing. The pCMV-FLAG-RECQ5 construct was generated during our previous study[15]. PIPA and PIP-LA RECQ5 mutants were created by site-directed mutagenesis by using pCMV-FLAG-RECQ5 as the template.

**Antibodies**. Mouse α-PCNA PC10 (sc-56, 1:5000), goat α-SRSF1 (sc-10254; 1:1000), rabbit α-U2AF65 (sc-48804; 1:5000), mouse α-tubulin (sc-8035; 1:3,000), rabbit α-H3 (sc-10809), rabbit α-DNA pol δ (sc-10784; 1:5000), rabbit α-LAMIN A/C (sc-20681, 1:5,000), rabbit α-Myc (sc-789; 1:1000), mouse α-His (sc-8036; 1:1000), rabbit α-HA (sc-805; 1:1000), goat α-actin (sc-1616; 1:1000), and mouse α-RNAPII A10 (sc-17798; 1:1000) were from Santa Cruz Biotechnology. Mouse α-RNAPII phospho-CTD (phospho S5; 4H8; C49196; 1:5000) was from Lifespan Biosciences Inc. Rabbit α-MCM7 (ab52489; 1:5000), rabbit α-Ub (ab7780; 1:1000), rabbit α-histone H3K9me3 (ab8898; 1:1000), rabbit α-γH2AX for immunofluorescence (ab11174; 1:500), rabbit α-SUMO2/3 (ab3742; 1:1000), and mouse α-RNAPII phospho-CTD (phospho S2; H5; ab24758; 1:5,000) were from Abcam. Goat α-MCM2 (A300-122A; 1:2000) and rabbit α-SPT16 (A302-492A; 1000) were from Bethyl Laboratories. Rabbit α-Histone H4 (#2592; 1:1000) was from Cell Signaling. Mouse α-NWSHPQFEK tag (StrepII tag; A01732; 1:3000) was from GeneScript. Mouse α-p84 (GTX70220; 1:5,000) was from GeneTex. Rabbit α-FLAG (F7425; 1:5,000) was from Sigma-Aldrich. Mouse α-γH2AX (NP002096) for ChIP was from EMD Millipore. Rabbit α-CAF1A (NB100-74608; 1:1,000) was from Novus Biologicals. Mouse α-histone H3K9me2 (39683; 1:1000) was from Active Motif. Fluor® 488 AffiniPure goat α-rabbit IgG (H+L) (111-545-144; 1:200) and goat α-rat DyLight 488-conjugated IgG (112-485-167; 1:200) were from Jackson Immunoresearch. Rat α-BrdU (MCA2060T; 1:200) was from AbD Serotec. Mouse α-IdU (347580; 1:200) was from BD, and rat α-CldU (MCM2020T; 1:200) was from BioRad. Rabbit α-PCNA (1:2000) was kindly provided by Dr. Robert Hickey (City of Hope). Rabbit α-RECQ5 (1:3000) was generated during our previous study[17].

**Cell culture and cell cycle synchronization**. HEK293T, A549, and HeLa cells were cultured in DMEM medium supplemented with 10% v/v fetal bovine serum (FBS) or Serum Plus II (Sigma) and streptomycin/penicillin (100 U ml⁻¹). HeLa cells were also supplemented with sodium pyruvate. HCT116 cells were cultured in McCoy's 5α medium supplemented with 10% FBS and streptomycin/penicillin. All cell lines were confirmed free of mycoplasma contamination. RECQ5 stealth siRNA 5′-UAG ACU UGG CAA UAUUCC AAU GGG C-3′ was purchased from Invitrogen. Plasmids and siRNAs were transfected using the Continuum™ Transfection

Reagent (GEMINI) according to the manufacturer's protocol. For cell cycle analysis, the cells were synchronized at the $G_2/M$ phase with 50 ng/ml nocodazole in complete medium for 20 h, and were released by washing twice with complete DMEM medium[63]. For CPT (5 μM) and DRB (100 μM) treatments, cells were exposed to the indicated agent for 2 h. For UV treatment, cells in log phase were irradiated with 30 J m$^{-2}$ UV, followed by 1 h of incubation before harvest.

**Cell fractionation and protein purification from human cells**. Cells were lysed (30 min, on ice) in 3 volumes of cytoplasmic buffer (10 mM Tris–HCl pH 7.5, 0.34 M sucrose, 3 mM CaCl$_2$, 2 mM MgCl$_2$, 0.1 mM EDTA, 1 mM DTT, 0.5% NP40, 40 mM NEM) containing protease and phosphatase inhibitors. The nuclear pellet was collected by centrifugation (2400 × g, 5 min). Nuclei were then resuspended in 3 volumes of nuclear buffer (20 mM HEPES pH 7.5, 1.5 mM MgCl$_2$, 1 mM EDTA, 150 mM KCl, 0.1% NP40, 1 mM DTT, 10% Glycerol) and homogenized with a 21G1/2 needle. The intact chromatin pellet was collected after centrifugation (18,000 × g, 30 min). To obtain the CB fraction, the chromatin pellet was incubated with 2 volumes of nuclease buffer (20 mM HEPES pH 7.5, 1.5 mM MgCl$_2$, 1 mM EDTA, 150 mM KCl, 10% Glycerol, 0.5 U μl$^{-1}$ benzonase) overnight at 4 °C, and the supernatant was collected as the CB fraction. Alternatively, to obtain separate CB:RNA+ and CB:RNA− fractions[21], the chromatin pellet was first incubated with RNase A in RNase A buffer (50 mM Tris–HCl pH 8.0, 10 mM EDTA, 150 mM NaCl, RNase A 10 μg ml$^{-1}$) for at least 2 h to overnight at 4 °C. The supernatant was collected as the CB:RNA+ fraction. The remaining pellet was then digested with benzonase for at least 2 h to overnight at 4 °C in the nuclease buffer, and the solubilized proteins were collected as the CB:RNA− fraction. To immunopurify FLAG-tagged protein complexes, chromatin extracts were incubated overnight with M2-agarose (Sigma) at 4 °C. After binding of the protein complexes, beads were washed extensively with FLAG-A binding buffer (10 mM HEPES pH7.9, 1.5 mM MgCl$_2$, 0.3 M NaCl, 10 mM KCl, 0.2% Triton X-100, 10% glycerol). The purified FLAG-tagged protein complexes were eluted by using either SDS loading buffer or FLAG elution A buffer (10 mM HEPES 7.9, 0.2 M NaCl, 0.2 mM EDTA, 0.05% Triton-X, 0.3 mg ml$^{-1}$ FLAG peptide, 10% glycerol)[21,63]. To purify His-Myc-PCNA from the CB:RNA+ and CB:RNA− fractions under denaturing conditions, the fractions were diluted with 10 volumes of 6 M Guanidinium-HCl buffer, and proteins were purified using Ni-NTA. All mass spectrometry analyses were conducted by the Taplin Mass Spectrometry Facility at Harvard University.

**Bacterial protein expression and purification**. StrepII-PCNA, StrepII-S2-PCNA fusion, and StrepII-GFP were overexpressed in *E. coli* BL21(DE3) for 4 h with 0.5 mM isopropyl β-D-1-thiogalactopyranoside (IPTG) at 37 °C. Cells were pelleted and resuspended in lysis buffer (50 mM Tris pH 8.0, 0.5 M NaCl, 0.5% Triton-X-100, 15% glycerol, protease inhibitor cocktail Complete™ [Roche]). Cells were lysed and debris was removed by centrifugation at 20,000 × g for 30 min. The supernatant was incubated with Strep-Tactin superflow beads (IBA Lifesciences) on a rocking platform for 4 h at 4 °C. After washing 10× with 50 volumes of lysis buffer, bound proteins were eluted with elution buffer (1 × PBS, 2.5 mM desthiobiotin, 10% glycerol) and dialyzed against storage buffer. Fractions containing the protein were dialyzed against Buffer B (50 mM Tris pH 8.0, 0.25 M NaCl, 10% glycerol, 1 mM DTT, 0.5 mM EDTA) and stored at −80 °C. To purify His-SUMO2 for in vitro SUMOylation reactions, expression of His-SUMO2 was induced for 4 h with 0.1 mM IPTG at 37 °C. Cells were pelleted and resuspended in Buffer C (50 mM Tris pH 8.0, 0.3 M NaCl, 0.5 mM EDTA, 1 mM DTT, 10% glycerol, 0.5% Triton-X-100, protease inhibitor cocktail Complete™ [Roche]). His-SUMO2 proteins were bound to Ni-NTA affinity resin (Qiagen), eluted with Buffer C containing 200 mM imidazole and dialyzed against Buffer B (50 mM Tris pH 8.0, 0.25 M NaCl, 10% glycerol, 1 mM DTT, 0.5 mM EDTA) before storage at −80 °C. His-CAF1A-FLAG was overexpressed in *E. coli* BL21(DE3). Cells were grown at 37 °C to a cell density OD600 ~0.4, followed by induction at 16 °C with 0.1 mM isopropyl-β-D-thio-galactoside (IPTG) for overnight. Cells were harvested by centrifugation and lysed in lysis buffer (50 mM potassium phosphate, pH 8.0, 10% glycerol, 300 mM KCl, 0.5% Triton-X100, 5 mM β-mercaptoethanol, 1 × protease inhibitor cocktail (Roche), 1 mM PMSF, and 0.2 mg ml$^{-1}$ lysozyme), followed by sonication. The supernatant was clarified by centrifugation and applied to Ni-NTA column, washed with Buffer B (50 mM potassium phosphate, pH 8.0, 10% glycerol, 300 mM KCl, 0.5% Triton-X100, 5 mM β-mercaptoethanol), and the His-tagged proteins were eluted with Buffer B containing 1 M imidazole. The eluate was dialyzed against Buffer D (50 mM Tris-HCl, pH 7.6, 10% glycerol, 300 mM KCl) and incubated with α-FLAG M2 beads (Sigma) overnight at 4 °C. The M2 bound proteins were washed with and stored in Buffer E (50 mM Tris-HCl, pH 7.6, 10% glycerol, 300 mM KCl, 0.2% Triton-X100, 1 mM EDTA).

**In vitro SUMOylation and pull-down assays**. To generate SUMOylated StrepII-PCNA for Fig. 4b, in vitro SUMOylation was carried out at 37 °C for 4 h using a SUMOylation kit (Enzo). SUMOylation reactions contained 1 mg StrepII-PCNA and 1 mg His-SUMO2. Upon completion, SUMOylation reactions were diluted with 10 volumes of Ni-NTA binding buffer (50 mM Tris pH 8.0, 0.3 M NaCl, 10% glycerol, 0.1% Triton-X-100, 5 mM imidazole) and incubated with Ni-NTA beads overnight at 4 °C. After washing 10× with 50 volumes of binding buffer, the His-SUMO2-conjugated StrepII-PCNA proteins were eluted with PBS containing 500

mM imidazole. Eluted fractions were diluted with 10 volumes of PBS and incubated with Strep-Tactin beads for 2 h at 4 °C, followed by 10× washes with PBS containing 0.1% Triton-X-100. For protein-protein interactions shown in Fig. 4b, 200 μl of the CB fraction was added to Strep-Tactin beads bound with the indicated Strep-tagged proteins and incubated for 4 h at 4 °C. Unbound proteins were removed by extensive wash with FLAG-binding buffer (10 mM HEPES 7.9, 1.5 mM MgCl$_2$, 250 mM NaCl, 0.1% Triton-X-100, 10% Glycerol), and bound proteins were analyzed by western blots. Similar pull-down was performed to generate data presented in Fig. 4e, except that the CB fractions were prepared from HEK293T-cells with or without FLAG-SSRP1 expression and Strep-tagged S2-PCNA fusion was used. For Fig. 4c, e, FLAG-CAF1A was first purified from either HEK293T cells (Fig. 4c) or *E. coli* (Fig. 4d) and bound to FLAG M2 agarose beads (Sigma). The FLAG-CAF1A-containing beads were then incubated with either StrepII-PCNA or StrepII-S2-PCNA purified from *E. coli*. The unbound proteins were removed and the bound proteins were analyzed, as described above.

**In vitro histone deposition assays**. Pre-assembled H3-H4 tetramer and H2A-H2B dimer were purchased from New England Biolabs. Non-assembled core histones were purified from nuclei of HeLa cells by acid extraction[39]. Briefly, HeLa cells were suspended (at a density of $10^7$ cells ml$^{-1}$) and lysed in PBS containing 0.5% Triton-X-100 and 2 mM phenylmethylsulfonyl fluoride (PMSF) for 10 min on ice. Intact nuclei were collected by centrifugation (1000 × g, 15 min) washed twice with lysis buffer and resuspended in 0.2 N HCl (density of $4 × 10^7$ cells ml$^{-1}$). Acid extraction of the histones was carried out overnight at 4 °C. Samples were then centrifuged (20,000 × g, 15 min, 4 °C), and the supernatants (which contained the histones) were collected and neutralized with 1 M Tris–Cl (pH 9.0). NaCl was added to a final concentration of 2 M. For immunodepletion of CAF1, 5 μg non-specific rabbit IgG or rabbit α-CAF1A antibody was incubated with 100 μg of cytoplasmic extracts prepared from SUMO2-PCNA overexpressing HEK293T cells at 4 °C for 12 h. Protein A/G magnetic beads (ThermoFisher Scientific, 50 μl, 50%) were added to each reaction, which was incubated at 4 °C for an additional 4 h before the supernatant was collected for analysis.

For DNA supercoiling assays to detect nucleosome assembly, purified DNA plasmids (200 ng) were relaxed by TOPO I (5 units per reaction, Invitrogen) at 37 °C for 1 h in reaction buffer (10 mM Tris, pH 7.5, 100 mM NaCl, 2 mM MgCl$_2$, 0.5 mM DTT, 3 mM ATP, 100 μg ml$^{-1}$ BSA) in a total volume of 15 μl. In a separate reaction, purified histones (800 ng) from HeLa cells or the pre-assembled H3-H4 tetramer (0.5 μM) or H2A-H2B dimer (1 μM) were incubated (30 min, 37 °C, in reaction buffer) with cytoplasmic extracts (6 μg each or otherwise indicated) obtained from hypotonic lysis of HEK293T cells transfected with the control vector, PCNA (WT), PCNA (KR), or SUMO-PCNA (KR). Hypotonic lysis was carried out on ice for 30 min used hypotonic buffer (10 mM HEPES pH7.5, 60 mM KCl, 1 mM EDTA, 5 mM DTT) containing PMSF and proteinase inhibitors. Nucleosome assembly was initiated by combining the relaxed DNA with the histone and cytoplasmic extract reactions and incubating for 1 h at 37 °C. To stop the reaction, an equal volume of stop buffer (20 mM EDTA, 1% SDS, and 200 μg ml$^{-1}$ proteinase K) was added, followed by incubation at 37 °C for 30 min, extraction with phenol/chloroform/isoamyl alcohol (25:24:1), and precipitation with ethanol. The purified DNA was separated by electrophoresis using a 1.2% agarose gel and visualized by staining with ethidium bromide (EtBr).

**Immunofluorescence microscopy and MNase sensitivity assay**. Immunofluorescence microscopy to detect γH2AX foci was performed by growing cells on coverslips for 1–2 days, washed twice with PBS, fixed in 3.7% paraformaldehyde for 10 min at room temperature and permeabilized with 0.2% Triton-X-100 for 10 min on ice. The coverslips were washed three times with PBS, followed by the sequential incubations with rabbit anti-γH2AX antibody and Fluor® 488 AffiniPure goat α-rabbit IgG. For neutral comet assay, cells were resuspended in PBS at a density of $1 × 10^5$ cells ml$^{-1}$, fixed in low melting-point agarose, spread evenly onto a pre-coated slide, dried at 4 °C for 30 min and lysed in pre-chilled lysis solution (Trevigen, UK) for 30 min at 4 °C in the dark. Slides were then washed in 1 × TBE and electrophoresed at a constant voltage of 1 V cm$^{-1}$ for 30 min. Slides were immersed in 70% ethanol for 5 min, air dried, stained using SYBR green and air dried for at least 24 h before viewing with an fluorescent microscope. The comet tail moment was recorded by using Comet Score 2.0 software. For DNA fiber analysis, 48 h after transfecting HEK293T cells with the corresponding PCNA plasmids, cells were labeled with 10 μM IdU for 10 min, followed by 100 μM CldU for 20 min. 500 cells were placed on a silane-pre slide (Sigma) in SB buffer (200 mM Tris–HCl pH 7.4, 50 mM EDTA, 0.5% SDS). The slide was tilted to spread the DNA, and the DNA was fixed by methanol:acetic acid (3:1) mixture, followed by denaturation using 2.5 N HCl. The slide was incubated with α-CldU and α-IdU antibodies to visualize CldU and IdU incorporation, and microscopy was conducted using a Zeiss Axio Observer. Fiber length was measured based on a conversion factor of 1 μm to 2.59 kb. The MNase assay was carried out by lysing the cells with ice-cold NP-40 lysis buffer (10 mM Tris–HCl pH 7.4, 10 mM NaCl, 3 mM MgCl$_2$, 0.5% NP-40, 0.15 mM spermine and 0.5 mM spermidine) on ice for 5 min. The nuclei were collected by centrifugation, washed and resuspended in MNase digestion buffer (10 mM Tris–HCl pH 7.4, 15 mM NaCl, 60 mM KCl, 0.15 mM spermine and 0.5 mM spermidine and 1 mM CaCl$_2$). The MNase digestion was initiated by adding the appropriate amount of MNase, and the reaction was terminated by the addition of

0.1 V MNase stop buffer (100 mM EDTA, 10 mM EGTA pH 7.5, 3.5 mg ml$^{-1}$ proteinase K and 1% SDS). The digested DNA was separated on 1% agarose gel and visualized by EtBr staining. EU staining and transcription analysis was carried out using Click-iT RNA Alexa Fluor® 488 Imaging Kit (Invitrogen) according to the manufacturer's protocol.

**DRIP and ChIP**. For DRIP analysis, cells were lysed in SDS/Proteinase K buffer (50 mM Tris pH 8.0, 10 mM EDTA, 0.5% SDS and 300 µg ml$^{-1}$ proteinase K) at 37 °C overnight, followed by phenol/chloroform extraction and ethanol precipitation to purify nucleic acids. Nucleic acids were fragmented using the restriction enzymes XbaI and SacI with or without RNase H at 37 °C overnight. After phenol/chloroform extraction and ethanol precipitation, 4 µg of nucleic acids per sample were immunoprecipitated with 2.5 µg of S9.6 antibody in binding buffer (10 mM NaPO$_4$ pH 7.0, 140 mM NaCl, 0.05% Triton X-100) at 4 °C over night. The binding complexes were then incubated with Protein A agarose for 1 h at 4 °C and washed with binding buffer extensively. The bound nucleic acids were eluted with SDS/Proteinase K buffer at 50 °C for 1 h, and purified by phenol/chloroform extraction and ethanol precipitation. For RNAPIIo and γH2AX ChIP analyses, cells were fixed with 1% formaldehyde for 10 min at room temperature, and the reaction was stopped by the addition of 0.125 M glycine Cells were resuspended in buffer I (5 mM PIPES pH 8.0, 85 mM KCl, 0.5% NP-40), followed by homogenization using a Dounce homogenizer. Chromatin pellet was isolated by centrifugation at 5000 rpm for 5 min at 4 °C. The pellets were re-suspended in ChIP lysis buffer (1.0% SDS, 10 mM EDTA, 50 mM Tris pH8.0) plus protease inhibitors and chromatin was sheared by sonication to generate DNA fragments of <1 kb. Chromatin was diluted 10 times in ChIP Dilution Buffer (16.7 mM Tris pH 8.0, 0.01% SDS, 1.1% Triton X-100, 1.2 mM EDTA and 167 mM NaCl) plus protease inhibitor and pre-cleared with protein A/G beads (Thermo Scientific) for 1 h at 4 °C. Pre-cleared samples were incubated overnight at 4 °C with antibodies. For RNAPIIo ChIP, the beads were washed sequentially twice with low salt buffer A (0.1% SDS, 1.0% Triton X-100, 2 mM EDTA, 20 mM Tris pH8.0 and 0.15 M NaCl), high salt buffer A (0.1% SDS, 1.0% Triton X-100, 2 mM EDTA, 20 mM Tris pH 8.0 and 0.5 M NaCl), LiCl buffer A (0.25 M LiCl, 1.0% NP-40, 1% sodium deoxycholate, 1 mM EDTA and 10 mM Tris, pH 8.0) and TE buffer. The RNAPIIo–DNA complexes were eluted with 300 µl elution buffer (0.1 M sodium bicarbonate, 1.0% SDS) at room temperature for 15 min, reverse cross-linked by adding 20 µl of 5 M NaCl and incubated at 65 °C overnight. The DNA was digested with RNase A and proteinase K and purified by phenol-chloroform and ethanol precipitation. For γH2AX ChIP, beads were washed once in dialysis buffer (2 mM EDTA, 50 mM Tris pH 8, 0.2% Sarkosyl) and four times in wash buffer (100 mM Tris pH 8.8, 500 mM LiCl, 1% NP-40, 1% sodium deoxycholate). After washing, the beads were re-suspended in 200 µl of TE buffer, and formaldehyde crosslink was reversed in the presence of 0.5% SDS at 70 °C overnight. DNA was purified with phenol/chloroform and ethanol precipitation. HA-HA3.1 and HA-H3.3 ChIP experiments were performed without fixation. Briefly, isolated nuclei were resuspended in nuclear buffer (20 mM HEPES pH 7.9, 1.5 mM MgCl$_2$, 150 mM KCl, 1 mM EDTA, 1 mM DTT, 10% glycerol, and 0.1% NP-40 plus protease inhibitor), followed by sonication to generate DNA fragments of an average length of 500 bps. HA agarose (Sigma) was added to the sonicated nuclear extracts to immunoprecipitate HA-tagged proteins and associated DNA. The HA agarose beads were subsequently washed twice with each of the following buffers: low salt wash buffer (20 mM Tris–HCl pH 8.0, 150 mM NaCl, 0.01% SDS, 1.1% Triton-X-100, 1.1 mM EDTA), high salt wash buffer (20 mM Tris-HCl pH 8.0, 500 mM NaCl, 0.01% SDS, 1.1% Triton-X-100, 1.1 mM EDTA), LiCl wash buffer (20 mM Tris–HCl pH 8.0, 0.5 M LiCl, 1% NP-40, 1 mM EDTA, and 1% deoxycholate) and TE buffer. The bound protein–DNA complexes were then eluted with elution buffer (1% SDS, 0.1 M NaHCO$_3$) and treated with Proteinase K at 45 °C for 2 h. qPCR primers against WWOX and IMMP2L and ACTBUS were previously described[3,21]. qPCR was performed using an ABI 7500 Fast real time PCR system and SYBR Green. Enrichment was calculated using the comparative Ct method. Each value represents the average value±standard deviation calculated from triplicate qPCR reactions per one representative experiment.

**Data availability**. All data generated or analyzed during this study are included in this published article (and its supplementary information files) and can be obtained from the authors upon reasonable request.

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

## Acknowledgements

We thank Dr. Keely Walker and Dr. Kerin Higa for their comments and expert editing of this manuscript. This work was supported by NIH R01 CA151245 to Y.L., NIH R50 CA211397 to L.Z., and NIH R01 CA073764 to B.H.S.

## Author contributions

M.L. and Y.L. conceived the study. M.L. and X.X. performed the majority of the experiments. C.-W.C. carried out the H3 ChIP, DRIP, and cell cycle experiments. L.Z. performed the supercoiling assay for histone deposition. M.L. and Y.L. analyzed and interpreted the data. Y.L. wrote the manuscript. B.H.S. helped to interpret the data and proofread the manuscript.

## Additional information

**Competing interests:** The authors declare no competing interests.

