## [Peer Review File · Nature Communications]

Reviewers' comments:

Reviewer #1 (Remarks to the Author):

Li et al. demonstrate that SUMO2 conjugation of PCNA enhances replication fork progression and destabilizes RNAP from chromatin. They identify a histone exchange mechanism involving H3.1 and H3.3 that is facilitated by SUMO2-PCNA, reducing chromatin accessibility and RNAP occupancy. Finally, they demonstrate a role for RecQ in modulating SUMO2-PCNA levels in the cells. The findings are important and interesting. Overall, I found the experiments to be nicely controlled, and the claims made by the authors to be supported by the data. There are a couple of experiments that I think would strengthen their argument, and really prove that their model is correct. I also have some issues with the writing:

1) The authors need to perform the ChIP experiments in non-replicating cells as a control. They claim that their observations are replication-dependent but do not demonstrate this critical point. This is particularly important in the context of RNAPII occupancy experiments. Is the effect really replication-dependent?

2) It seems that a DNase treatment of the protein-protein interaction samples and mass spec samples would be important to do. Some of the factors are suggested to be chromatin associated and therefore should be lost if the samples are DNase treated. I wanted to see this particular controls in several places including the results described on page 11 through 13.

3) This may be a relatively philosophical point. However, I think its an important one. The authors generally refer to this mechanism as a mechanism of "avoidance" or a mechanism that "minimizes" conflicts. This is not correct. What occurs at these sites, as they describes, follows the conflict. In other words, first the fork arrives at the transcribed regions and then this triggers the SUMO2-PCNA mediated mechanism that helps actually RESOLVE the conflict. The conflicts are unavoidable for the most part, and these mechanisms generally are resolution mechanisms. I recommend the authors change the writing to reflect that. If the SUMO2-PCNA mediated mechanism is activated upon the arrival of the fork at the transcribed regions, then this is not an avoidance strategy. In the same lines, the authors suggest in the intro that the fork disassembles before arriving at the conflict site. This doesn't make any sense – how would the fork know there are conflicts that will be happening in the future? In my opinion, this type of language must be taken out of the paper.

4) The English usage and grammar in this paper needs a lot of help. In particular, the abstract and intro are difficult to follow because of language problems. The manuscript would benefit quite a bit from edits to the language by a native speaker.

5) The authors should explain the mechanisms of inhibition by the drugs used in the experiments. For example, how does DRB work? These explanations will generate some clarity for non-experts.

6) On page 7, the last part of the first paragraph referring to Fig 1f and Supp fig 2c seems very much out of place and should be moved to a later section in the paper.

Reviewer #2 (Remarks to the Author):

In this study, Li et al. investigate the molecular mechanism underlying RNAPII dissociation from chromatin during replication in human cells. They uncover a transcription-dependent SUMO2-conjugation of PCNA on K164 during S-phase, which is dependent on the RNAPII-associated RECQ5 helicase. They demonstrate that SUMO2-conjugated PCNA destabilizes RNAPII from chromatin and enhances replication progression. In addition, they show that SUMO2-PCNA interacts with the histone chaperones CAF1 and FACT, which is accompanied by increased H3.1 and reduced H3.3 occupancy at common fragile sites. Finally, they provide evidence that SUMO2-PCNA minimizes transcription-induced DNA damage at common fragile sites.

This is an interesting and topical study that convincingly demonstrates the transcription-dependent SUMO2-conjugation of PCNA during S phase and then investigates the consequences of SUMO2-

PCNA on replication, transcription, chromatin assembly and genome instability. However, connections between these events are not established and many of the authors' conclusions are not fully supported by the data provided. Furthermore, mechanistic insights into the proposed model are still lacking, important controls are missing and some aspects require clarification (as detailed below). Therefore, additional experiments are necessary to improve the manuscript before it can be considered for publication in Nature Communications.

Major comments:

1- The analysis is performed only in HEK293 cells. The authors should confirm some of their key findings in other cell types, preferentially primary cells.

2- The authors show that the levels of chromatin-bound RNAPII_o are reduced in cells over-expressing SUMO2-PCNA (Fig 2c). They should monitor RNAPII_o levels in WCE to make sure that RNAPII_o indeed dissociates from chromatin and that the total RNAPII_o levels are unchanged. To strengthen their findings, they could also monitor the effect of SUMO2-PCNA on the overall levels of transcription (by EU staining for example).

3- The authors show that SUMO2-PCNA enhances replication fork progression (Fig 2d-e). To support their findings, the speed of S phase progression could be monitored by FACS. .

4- Analyses at common fragile sites are performed on only 2 loci (Fig 3, Fig 6e, Suppl Fig 5). A more extensive analysis is required to draw firm conclusions.

5- Contrary to what is stated (p.13), the authors do not directly address whether SUMO2-PCNA facilitates histone turnover and increases repressive chromatin. To draw such conclusion, histone modifications associated with transcription repression should be examined. Note that the histone variant H3.3 does not only mark open and transcriptionally active chromatin (p.4, p.19), as it is also deposited in heterochromatin and carries repressive histone marks (Elsaesser, Nature 2015; Udagama NAR 2015; Goldberg, Cell, 2010; Lewis, PNAS 2010; Drane, Genes Dev 2010; Wong, Genome Res 2009; Banaszynski, Cell 2013). The authors should take this into account.

6- To claim that the greater chromatin assembly activity is "driven by CAF-1" (p.14), CAF-1 should be depleted from the extracts used to perform the supercoiling assay otherwise there is no evidence for a "CAF-1-mediated histone deposition activity" (p.14). Also, is CAF-1 involved in the decreased chromatin accessibility observed in the MNase assay (Fig 5d)?

7- The authors state that SUMO2-PCNA both enhances replication fork progression and destabilizes RNAPII binding "as a consequence" of CAF-1 dependent deposition of H3.1 (p.4). Is CAF-1 activity or H3.1 deposition required to dislodge RNAPII, to generate repressive chromatin, and/or to suppress transcription-induced damage?

8- gH2AX is not a direct readout of DSBs. TUNEL or comet assays should be performed to measure DSBs. Moreover, when investigating DNA damage arising from replication-transcription conflicts, R-loops must be examined as done by the authors in a previous study (Li, Nat Comm 2015).

9- The authors put forward RECQ5 as "a key enzyme for PCNA SUMO2 conjugation" (p.21, Fig 7). From their data, RECQ5 mediates the interaction between PCNA and RNAPII and it does not tell much about the actual mechanism of PCNA sumoylation as RECQ5 is not a SUMO ligase. It would thus be important to identify the responsible SUMO ligase. The lack of PCNA sumoylation upon RECQ5 knock-down could also be explained by an indirect effect on transcription as PCNA sumoylation is transcription-dependent. The authors should examine the effect of RECQ5 loss and gain of function on RNAPII occupancy and transcription.

10- In the discussion, the authors should consider the orientation of transcription-replication

conflicts - co-directional or head-on – in light of a recent the study by the Cimprich lab (Hamperl et al, Cell 2017). In addition, a working model could be included in Fig. 7.

11- A number of loading controls are missing: Fig 2e, Fig 6a, Fig 7c. Controls for chromatin-bound fraction are missing in Fig 2c (also in suppl Fig 3c). Statistical analyses are missing in some panels of Fig 3c, Fig 5c, Fig 6. A positive control is missing for SUMO1 detection (Fig. 1e). Suppl Fig 3 does not provide direct evidence for ubiquitinated SPT16.

Minor points:

-The identification of K164 as the SUMO2-conjugated residue (Suppl. Fig. 2c) could be moved to the main figure.

- The average from independent CHIP experiments should be presented (Fig. 3, 5, 6, Suppl Fig.5) instead of that from internal replicates. Alternatively, the result of an independent experiment should be included in the supplementary data.

-The scheme in Fig. 2a could be more detailed and include the position of the mutated lysine.

-The authors should explain how gH2AX positive cells are defined (Fig. 6b): minimum intensity, minimum number of foci? The authors should comment on the fact that S2-KR PCNA induces more damage than WT PCNA.

-The authors should explain why they use CPT in the experiments shown on Fig. 7a-b. They should also comment on the presence of 3 different size bands for RECQ5 (Fig. 7c). Figure 7d should show a longer exposure like in panel 7c to conclude that RECQ5 interacts only with non-modified PCNA.

- Suppl Fig 2b: it is unclear that SUMO2 is identified by MS on the given protein sequence.

Text editing:

- For non-expert readers, the authors should briefly define RNAPII α and RNAPII β forms.

- p. 6: The last two sentences are unclear. I guess that the point the authors want to make is that PCNA* modification is distinct from PCNA ubiquitylation induced by UV damage.

- p.11: The first sentence should read "To determine how SUMO2-PCNA attenuates RNAPII occupancy during replication...". "MutS" should be replaced by "mismatch repair factors". There is no catalytic subunit in CAF-1.

- p.14: Only MCM2 has been shown to promote histone recycling at the replication fork.

- p.15: It should read "with diminished H3.3 occupancy in chromatin at the examined loci".

- The authors should rephrase their conclusion about PIP-LA mutation not affecting the interaction between RECQ5 and RNAPII (p.18) as RECQ5-PIP-LA mutant pulls down more RNAPII than WT RECQ5 (Fig 7h).

- p.21: The following sentence "However, similar in vitro analysis suggests that PIAS1 is not the SUMO2 E3 ligase for PCNA" should refer to a publication or to data not shown.

Reviewer #3 (Remarks to the Author):

This paper describes the finding that SUMOylated PCNA – or at least a fusion of SUMO and PCNA is able to reduce replication-transcription conflicts.

It makes the case that PCNA is modified in transcribed regions in a manner that depends on

interaction with RECQ5. SUMOylated PCNA in turn interacts with histone chaperones FACT and CAF1 encouraging H3.1 deposition and reduced RNAPoIII interaction. This activity reduces transcription dependent DNA damage.

The story is timely and a good fit for Nature Comms. I like the robust nature of the investigation and the clarity of the report. However in my opinion there are a couple of approaches that need to be tightened before publication.

The RECQ5-L mutant data is good, showing a link to PCNA SUMOylation, but it is not yet tied into the system in an entirely satisfactory way. Critically this is needed to confirm the RECQ5 link. Does complementation of RECQ5 depletion with this mutant result in transcription-mediated double-strand breaks, (slowed forks over transcribed regions?) and in RNA-poll lingering at fragile sites? Most significantly can its impact be over-come by expression of PCNA-SUMO2 (and not by PCNA-Ub)?

While there are a great many questions this report raises I feel two are particularly out-standing. How does SUMO-PCNA improve FACT/CAF1A recruitment and how is RECQ5 responsible for SUMOylation of PCNA? How RECQ5 functions is clearly a story for another report but I feel a little more on the direct role of SUMOylated PCNA would flesh out the model considerably.

With the model available this is relatively straight forward as mutations can be introduced into SUMO2 to prevent SIM interaction and/or to make the protein more SUMO1-like to address the role for a SIM from the PCNA point of view. Ideally expression of a SIM mutant of FACT/CAF1A would complement the approach.

The idea that PIP-box + SIM makes a module for interacting with replication machinery in actively transcription regions is an exciting one. What is the proposed explanation for SUMO2, rather than SUMO1 involvement?

I look forward to seeing the results of these experiments to finish this excellent report.

Reviewer #1 (Remarks to the Author):

Li et al. demonstrate that SUMO2 conjugation of PCNA enhances replication fork progression and destabilizes RNAP from chromatin. They identify a histone exchange mechanism involving H3.1 and H3.3 that is facilitated by SUMO2-PCNA, reducing chromatin accessibility and RNAP occupancy. Finally, they demonstrate a role for RecQ in modulating SUMO2-PCNA levels in the cells. The findings are important and interesting. Overall, I found the experiments to be nicely

controlled, and the claims made by the authors to be supported by the data. There are a couple of experiments that I think would strengthen their argument, and really prove that their model is correct. I also have some issues with the writing:

(1) The authors need to perform the ChIP experiments in non-replicating cells as a control. They claim that their observations are replication-dependent but do not demonstrate this critical point. This is particularly important in the context of RNAPII_o occupancy experiments. Is the effect really replication-dependent?

Based on the reviewer's suggestion, we conducted RNAPII_o ChIP in PCNA WT-, KR-, or S2-KR-overexpressing cells with or without nocodazole treatment to reduce the percentage of cells in S-phase. We found that nocodazole treatment, which reduced the S-phase population from 32-34% to 20-22%, proportionally reduced the number of RNAPII_o molecules associated with CFS in PCNA WT- and KR-overexpressing cells. This is consistent with the literature that CFSs are primarily transcribed during S-phase. On the other hand, because CFS-associated RNAPII_o was already reduced by S2-KR overexpression, the decrease in the S-phase population by nocodazole treatment had a minimal effect on RNAPII_o levels at CFS in these cells. This result is now included in Suppl Fig. 4d-e.

(2) It seems that a DNase treatment of the protein-protein interaction samples and mass spec samples would be important to do. Some of the factors are suggested to be chromatin associated and therefore should be lost if the samples are DNase treated. I wanted to see this particular controls in several places including the results described on page 11 through 13.

We have clarified in the text (pp.10-11) that the chromatin-bound (CB) fractions were prepared by digesting the chromatin pellets with benzonase, a potent nuclease that digests both DNA and RNA, to solubilize proteins from the chromatin pellets. The benzonase-treated, soluble chromatin fractions were free of nucleic acids, and these fractions were used for the subsequent immuno-purification of protein complexes for mass spec and western blot analyses. This experimental approach can be found in the Methods section (p.26).

(3) This may be a relatively philosophical point. However, I think its an important one. The authors generally refer to this mechanism as a mechanism of "avoidance" or a mechanism that "minimizes" conflicts. This is not correct. What occurs at these sites, as they describes, follows the conflict. In other words, first the fork arrives at the transcribed regions and then this triggers the SUMO2-PCNA mediated mechanism that helps actually RESOLVE the conflict. The conflicts are unavoidable for the most part, and these mechanisms generally are resolution mechanisms. I recommend the authors change the writing to reflect that. If the SUMO2-PCNA mediated mechanism is activated upon the arrival of the fork at the transcribed regions, then this is not an avoidance strategy. In the same lines, the authors suggest in the intro that the fork disassembles before arriving at the conflict site. This doesn't make any sense – how would the fork know there are conflicts that will be happening in the future? In my opinion, this type of language must be taken out of the paper.

We agree with the reviewer's point, and we have replaced "avoid" with "resolve" throughout the text. We also revised the title to "SUMO2 conjugation of PCNA facilitates chromatin remodeling to resolve transcription-replication conflicts"

(4) The English usage and grammar in this paper needs a lot of help. In particular, the abstract and intro are difficult to follow because of language problems. The manuscript would benefit quite a bit from edits to the language by a native speaker.

We thank the reviewer for this suggestion. We have had our professional scientific editing team help us revise this manuscript. That being said, we are more than happy to further edit the text as the reviewers and/or editor see fit.

(5) The authors should explain the mechanisms of inhibition by the drugs used in the experiments. For example, how does DRB work? These explanations will generate some clarity for non-experts.

We have clarified in the text (p. 5) that DRB blocks the phosphorylation activation of RNAPII and NEM inhibits deubiquitination (p.6). We have also explained the use of UV and CPT to induce DNA damage and PCNA ubiquitination (p. 6, 17).

(6) On page 7, the last part of the first paragraph referring to Fig 1f and Supp fig 2c seems very much out of place and should be moved to a later section in the paper.

Fig. 1f provides further validation for the conclusion drawn from Fig. 1e that PCNA in the CB:RNA+ fraction is conjugated with SUMO2 but not SUMO1. Therefore, we feel that it is reasonable to place Fig. 1f immediately after Fig. 1e. We have also moved Suppl Fig 2c to Fig. 1g, as we think that it is a logical follow-up, identifying lysine 164 as the residue being modified by the SUMO2 conjugation established in Fig. 1e-f.

Reviewer #2 (Remarks to the Author):

In this study, Li et al. investigate the molecular mechanism underlying RNAPII dissociation from chromatin during replication in human cells. They uncover a transcription-dependent SUMO2-conjugation of PCNA on K164 during S-phase, which is dependent on the RNAPII-associated RECQ5 helicase. They demonstrate that SUMO2-conjugated PCNA destabilizes RNAPII from chromatin and enhances replication progression. In addition, they show that SUMO2-PCNA interacts with the histone chaperones CAF1 and FACT, which is accompanied by increased H3.1 and reduced H3.3 occupancy at common fragile sites. Finally, they provide evidence that SUMO2-PCNA minimizes transcription-induced DNA damage at common fragile sites.

This is an interesting and topical study that convincingly demonstrates the transcription-dependent SUMO2-conjugation of PCNA during S phase and then investigates the consequences of SUMO2-PCNA on replication, transcription, chromatin assembly and genome instability. However, connections between these events are not established and many of the authors' conclusions are not fully supported by the data provided. Furthermore, mechanistic insights into the proposed model are still lacking, important controls are missing and some aspects require clarification (as detailed below). Therefore, additional experiments are necessary to improve the manuscript before it can be considered for publication in Nature Communications.

Major comments:

(1) The analysis is performed only in HEK293 cells. The authors should confirm some of their key findings in other cell types, preferentially primary cells.

We attempted to conduct these experiments in human primary fibroblasts, but the transfection efficiency was not sufficient to achieve PCNA overexpression in these cells. Therefore, we chose to repeat the key experiments in additional human cell lines: HeLa, HCT116, and A549. All of these cell lines are commonly used in the study of cellular and molecular mechanisms of transcription and replication. These key experiments, which supported our previous conclusions, are shown in Suppl Fig. 1b, 4b, 4c, 5c, 5d, 7b, 8a, and 8b.

(2) The authors show that the levels of chromatin-bound RNAPII_o are reduced in cells over-expressing SUMO2-PCNA (Fig 2c). They should monitor RNAPII_o levels in WCE to make sure that RNAPII_o indeed dissociates from chromatin and that the total RNAPII_o levels are

unchanged. To strengthen their findings, they could also monitor the effect of SUMO2-PCNA on the overall levels of transcription (by EU staining for example).

We have now included western blot analysis of RNAPII in WCE to demonstrate that S2-KR overexpression does not affect total RNAPII levels in the cells (Fig. 2b). We also conducted EU staining and found that S2-KR-overexpressing cells exhibit reduced EU incorporation, consistent with our conclusion that SUMO2-PCNA limits RNAPII_o-chromatin association. This result is presented in Suppl. Fig. 3c.

(3) The authors show that SUMO2-PCNA enhances replication fork progression (Fig 2d-e). To support their findings, the speed of S phase progression could be monitored by FACS.

We performed FACS analysis to monitor the progression of S phase in cells overexpressing WT, KR, and S2-KR PCNA after release from nocodazole inhibition. We confirmed that all of the three cell lines were synchronized to G₂/M by nocodazole with similar efficiency. We found that WT- and S2-KR-overexpressing cells progressed through G₁ into early S-phase at similar speeds, but KR-overexpressing cells exhibited a delay in S-phase entry, as demonstrated by a higher G₁ population and a lower early S-phase population at the 7 h time point compared to WT- and S2-KR-overexpressing cells. Importantly, by 10 h after release from nocodazole, a greater number of S2-KR-overexpressing cells reached late S-phase than those overexpressing WT or KR. This result, shown in Suppl. Fig. 3a-b, is consistent with our conclusion that SUMO2-PCNA enhances fork progression either by increasing the DNA synthesis rate or by removing interference that might otherwise slow down DNA replication.

(4) Analyses at common fragile sites are performed on only 2 loci (Fig 3, Fig 6e, Suppl Fig 5). A more extensive analysis is required to draw firm conclusions.

We have included additional RNAPII and H3 ChIP analyses for FRA7I, FRA3B, and FRAXC loci in HEK293T cells (Suppl Fig. 4a and 7a). In addition, we performed ChIP analyses in HeLa and HCT116 cell lines (Suppl Fig. 4b, 4c, and 7b). These additional CFS loci and cell line analyses confirmed our conclusion that SUMO2-PCNA reduces RNAPII_o occupancy and increases histone H3.1 at CFSs.

(5) Contrary to what is stated (p.13), the authors do not directly address whether SUMO2-PCNA facilitates histone turnover and increases repressive chromatin. To draw such conclusion, histone modifications associated with transcription repression should be examined. Note that the histone variant H3.3 does not only mark open and transcriptionally active chromatin (p.4, p.19), as it is also deposited in heterochromatin and carries repressive histone marks (Elsaesser, Nature 2015; Udugama NAR 2015; Goldberg, Cell, 2010; Lewis, PNAS 2010; Drane, Genes Dev 2010; Wong, Genome Res 2009; Banaszynski, Cell 2013). The authors should take this into account.

We agree with the reviewers that although the literature has established that H3.3 is important for maintaining open chromatin structure for transcription, it has also been implicated in contributing to repressive heterochromatin structure. Therefore, we have edited this section. Because CAF1 has been shown to deposit H3.1 and promote H3K9me_{2/3} repressive histone marks, we conducted additional analyses of H3K9me_{2/3} marks and found that cells overexpressing either CAF1 or S2-KR contain higher levels of H3K9me_{2/3} marks compared to PCNA WT- and KR- overexpressing cells. These results, which are included in Fig. 5f-g, support our conclusion that SUMO2-PCNA facilitates repressive chromatin.

(6) To claim that the greater chromatin assembly activity is “driven by CAF-1” (p.14), CAF-1 should be depleted from the extracts used to perform the supercoiling assay otherwise there is no evidence for a “CAF-1-mediated histone deposition activity” (p.14). Also, is CAF-1 involved in the decreased chromatin accessibility observed in the MNase assay (Fig 5d)?

We agree with the reviewer that this is an important question. Our initial strategy to answer this question was to knockdown CAF1 in S2-KR overexpressing cells. However, although we were able to successfully deplete more than 90% of CAF1 by siRNA (ThermoScientific), we found that CAF1 knockdown led to a high percentage of apoptotic cells within three days of transfection. These apoptotic cells made it impossible to assess chromatin accessibility using MNase digestion due to genomic DNA degradation as a result of apoptosis. We also could not perform RNAPII ChIP analysis due to protein degradation of both replication and transcription factors. Therefore, we took an alternative approach by immuno-depleting CAF1 from cell extracts prepared from S2-KR-overexpressing cells. We found that CAF1 depletion abolishes the increased histone deposition activity in S2-KR-overexpressing cell extracts. This result is shown in Fig. 5b. We also demonstrated that overexpressing either CAF1 or S2-KR, but not PCNA KR or Ub-KR fusion, suppresses TRC-induced DSBs in RECQ5 knockdown cells, providing evidence for synergism between CAF1 and SUMO2-PCNA in preventing TRC-induced DSBs. These results are shown in Fig. 7j and Suppl Fig. 9.

(7) The authors state that SUMO2-PCNA both enhances replication fork progression and destabilizes RNAPII binding “as a consequence” of CAF-1 dependent deposition of H3.1 (p.4). Is CAF-1 activity or H3.1 deposition required to dislodge RNAPII, to generate repressive chromatin, and/or to suppress transcription-induced damage?

As mentioned above, it was not possible for us to evaluate CAF1 knockdown cells due to high levels of apoptosis. Therefore, we adapted alternative strategies and provided supporting evidence that the SUMO2-PCNA-mediated histone deposition activity is dependent on CAF1, and this function is important for suppressing transcription-induced DNA damage. Specifically, we show that:

- (a) immunodepletion of CAF1 from S2-KR overexpressing cell extracts abolishes the elevated histone deposition activity (Fig. 5b);
- (b) overexpressing either S2-KR or CAF1 in human cells increases repressive histone marks (Fig. 5f-g).
- (c) the high levels of TRC-induced DNA breaks in RECQ5 knockdown cells can be suppressed by overexpressing either S2-KR or CAF1, but not KR or Ub-KR (Fig. 7j; Suppl Fig. 9a-c).

(8) γ H2AX is not a direct readout of DSBs. TUNEL or comet assays should be performed to measure DSBs. Moreover, when investigating DNA damage arising from replication-transcription conflicts, R-loops must be examined as done by the authors in a previous study (Li, Nat Comm 2015).

We have now included native comet assays and found that PCNA KR overexpression substantially increases tail moments, indicative of a higher level of DSBs. These DSBs were partially suppressed by DRB treatment. Consistent with the γ H2AX analysis, we found that S2-KR-overexpressing cells contain fewer DSBs, and DRB has no effect on the level of tail moments in the S2-KR overexpressing cells. These results are now shown in Fig. 6d. We also conducted R-loop analysis and found that S2-KR overexpression does not influence R-loop levels in either CFS or non-CFS gene loci, suggesting that SUMO2-PCNA functions primarily to resolve transcription-replication collision rather than R-loops. These data are now shown in Fig. 6h.

(9) The authors put forward RECQ5 as “a key enzyme for PCNA SUMO2 conjugation” (p.21, Fig 7). From their data, RECQ5 mediates the interaction between PCNA and RNAPII and it does not tell much about the actual mechanism of PCNA sumoylation as RECQ5 is not a SUMO ligase. It would thus be important to identify the responsible SUMO ligase. The lack of PCNA sumoylation upon RECQ5 knock-down could also be explained by an indirect effect on transcription as PCNA sumoylation is transcription-dependent. The authors should examine the effect of RECQ5 loss and gain of function on RNAPII occupancy and transcription.

SUMO modifications of PCNA have been observed in human cells since 2012 (Moldovan, Mol Cell 2012; Gali, NAR 2012; Hendrick, Nat Comm 2015). However, to date, the identities of the SUMO1 and

SUMO2 E3 ligases for PCNA remain unknown, largely due to the fact that *in vitro* PCNA can be efficiently SUMOylated in the absence of a SUMO E3 ligase (Choe, Mol Cell 2017; Yang, J Vis Exp 2018), making it difficult to identify any SUMO E3 ligases. Therefore, although we agree with the reviewer that finding the SUMO E3 ligase specific for PCNA SUMO2 conjugation would be valuable, it is beyond the scope of this manuscript.

We also understand the reviewer's concern regarding the role of RECQ5. However, it is unlikely that the absence of SUMO2-PCNA in the RECQ5 knockdown cells is due to an indirect effect on transcription for the following reasons: If SUMO2-PCNA in RECQ5 depleted cells was an indirect effect of transcription, the SUMO2-PCNA modification would not be dependent on the RECQ5-PCNA interaction, as we have observed (Fig. 7g-h). In addition, as we have shown in Fig. 1b, one way to abolish the formation of SUMO2-PCNA is to inhibit transcription. However, we and others have shown that RECQ5 depletion does not inhibit transcription (Li, MCB 2011; Saponaro, Cell 2014). On the contrary, our study found that there are more RNAPII molecules accumulated on the chromatin in RECQ5-depleted cells (Li, MCB 2011). It is also worth mentioning that Saponaro and colleagues (Cell 2014) conducted a genome-wide analysis of RNAPII occupancy and showed that RNAPII is capable of progressing significantly faster through long genes, which include CFS-containing gene loci, in RECQ5-depleted cells than in the WT cells. This result indicates that RECQ5 is important for slowing down the progression of RNAPII through long genes. Our current study provides a mechanistic explanation for how this is achieved via promoting PCNA SUMO2 conjugation to allow the replication fork to temporarily suspend RNAPII transcription progression to resolve transcription-replication conflict.

(10) In the discussion, the authors should consider the orientation of transcription-replication conflicts - co-directional or head-on – in light of a recent the study by the Cimprich lab (Hamperl et al, Cell 2017). In addition, a working model could be included in Fig. 7.

Hamperl et al reported a very interesting observation regarding the ways in which the orientations of TRCs differentially regulate R-loop levels. However, the analysis conducted by Hamperl and colleagues was done using a plasmid-based system in cells, and the plasmids were most likely nucleosome free or contained nucleosome structures that were different from the genomic chromatin structure (Mladenova 2009). For this reason, it is difficult to speculate for how SUMO2-PCNA-mediated chromatin remodeling may contribute to plasmid-based TRCs. In addition, our studies did not find a significant change in the R-loop levels in the S2-KR-overexpressing cells, suggesting that SUMO2-PCNA does not contribute to the regulation of R-loops at least at the genomic DNA level. While our study cannot distinguish between a head-on or co-directional collision, we believe that SUMO2-PCNA-mediated chromatin remodeling to destabilize RNAPII can apply to both head-on and co-directional collisions. Due to word limitations, we were not able to include a detailed discussion on this subject, but we have stated in the discussion that our finding should apply to both orientations (p. 21). We also included a cartoon model in Fig. 8 in the revised manuscript.

(11) A number of loading controls are missing: Fig 2e, Fig 6a, Fig 7c. Controls for chromatin-bound fraction are missing in Fig 2c (also in suppl Fig 3c). Statistical analyses are missing in some panels of Fig 3c, Fig 5c (add p value), Fig 6. A positive control is missing for SUMO1 detection (Fig. 1e). Suppl Fig 3 does not provide direct evidence for ubiquitinated SPT16.

We have now included LAMIN A/C and tubulin as loading controls for Fig. 2c and Fig. 6a, respectively. Fig 2e does not contain western blots. The original Fig. 7c (now 7d, upper panels) showed the chromatin input for FLAG-RECQ5 IP (Fig. 7d, lower panels), and we used non-modified PCNA as loading control for the input. We have also included a new set of western blot analyses of the CB:RNA+ and CB:RNA- fractions prepared from cells with or without FLAG-RECQ5 overexpression; this set contains MCM7 as a chromatin fraction loading control (Fig. 7c).

We provided the p values for Fig. 5d (previously Fig. 5c). In addition, we have clarified in the figure legends that only the statistically significant p values are shown (for Fig. 3b, 3c, 5d, 5e, 6b, 6f, 6g & 6h;

Suppl Fig. 4a-c, 4e, 7a-b, 8a-b and 9c). The same SUMO1 antibody used in Fig. 1e has previously been used in Li et al, 2015 to demonstrate that topoisomerase 1 is conjugated with SUMO1. We have now included a positive control for this antibody using purified SUMO1-TOP1, shown in Supp Fig. 2c. We agree with the reviewer that Suppl Fig. 3c did not provide direct evidence for ub-SPT16 and have therefore removed it from the revised manuscript. Our new data indicate that the enhanced FACT association with SUMO2-PCNA occurs via the SSRP1 SIM motif. This result is presented in Fig. 4f.

Minor points:

(1) The identification of K164 as the SUMO2-conjugated residue (Suppl. Fig. 2c) could be moved to the main figure.

We have moved this figure to Fig 1g.

(2) The average from independent ChIP experiments should be presented (Fig. 3, 5, 6, Suppl Fig.5) instead of that from internal replicates. Alternatively, the result of an independent experiment should be included in the supplementary data.

All ChIP analyses were repeated at least three times to confirm our results, and a representative set for each experiment is presented in the paper. To demonstrate the reproducibility of the ChIP analysis, we have included additional ChIP analyses using other cell lines. These new ChIP data are included in Suppl Fig. 4b-c and 7b.

(3) The scheme in Fig. 2a could be more detailed and include the position of the mutated lysine.

We included a red line in the diagram to indicate the position of K164.

(4) The authors should explain how γ H2AX positive cells are defined (Fig. 6b): minimum intensity, minimum number of foci? The authors should comment on the fact that S2-KR PCNA induces more damage than WT PCNA.

We have clarified in the corresponding figure legend (p.45) that cells with 5 or more visible γ H2AX foci were considered γ H2AX-positive. We also stated in the text that the remaining DSBs found in the S2-KR-overexpressing cells compared to WT-overexpressing cells likely result from defects in other PCNA K164 modifications (p.16).

(5) The authors should explain why they use CPT in the experiments shown on Fig. 7a-b. They should also comment on the presence of 3 different size bands for RECQ5 (Fig. 7c). Figure 7d should show a longer exposure like in panel 7c to conclude that RECQ5 interacts only with non-modified PCNA.

DNA damaging agents, such as UV and CPT, have been shown to induce PCNA ubiquitination (Marko 2017). Therefore, we used both UV (Suppl Fig 1d) and CPT (Fig 7a-b) to demonstrate that the PCNA SUMO2 modification is not induced by DNA damage. We have included this information in the revised text (p.6 & p.17). In Fig 7d (old Fig. 7c), the RECQ5 band in lane 1 is endogenous RECQ5, which correlates with the minor band just below the major FLAG-RECQ5 band in lane 3. The slower mobility of FLAG-RECQ5 band in the CB:RNA- fraction compared to that in the CB:RNA+ fraction is likely due to gel distortion during electrophoresis, because the FLAG-RECQ5 proteins purified from these two fractions exhibit similar mobility on the SDS-PAGE gel (2nd panel from the bottom). As expected, the minor lower band that corresponds to the endogenous RECQ5 was not present after FLAG IP. The western blot image provided in Fig 7c (now Fig. 7d, bottom panel) is the longest exposure we had performed. At this exposure, the intensity of the non-modified PCNA band co-purified with RECQ5 is

comparable to the intensity of the non-modified PCNA band in the short exposure of the input (second panel from the top). However, even though we could clearly detect the presence of the SUMO2-PCNA band in the input (second panel from the top), we were not able to detect the presence of SUMO2-PCNA molecules in the IP sample (bottom panel). Consistent with this, we found that RECQ5 is co-immunoprecipitated by FLAG-PCNA WT or KR but not S2-KR (Fig. 4b). These results are consistent with our conclusion that RECQ5 preferentially interacts with non-modified, but not SUMO2-conjugated, PCNA.

(6) Suppl Fig 2b: it is unclear that SUMO2 is identified by MS on the given protein sequence.

We agree with the reviewer that the peptide sequence identified by mass spec ruled out the possibility of SUMO1 conjugation but cannot distinguish between SUMO2 and SUMO3. However, when we expressed His-SUMO1, 2 or 3 and performed Ni-NTA pull-down under denaturing conditions from CB:RNA+, we found that PCNA in the CB:RNA+ fraction is conjugated with SUMO2 but not SUMO3. We have edited the text to clarify this finding (pp. 6-7).

Text editing:

(1) For non-expert readers, the authors should briefly define RNAPII α and RNAPII β forms.

We have defined the two forms of RNAPII in the revised text (p.5).

(2) p. 6: The last two sentences are unclear. I guess that the point the authors want to make is that PCNA* modification is distinct from PCNA ubiquitylation induced by UV damage.

We thank the reviewer for the suggestion and have edited the text accordingly (p.6).

(3) p.11: The first sentence should read “To determine how SUMO2-PCNA attenuates RNAPII occupancy during replication...”. “MutS” should be replaced by “mismatch repair factors”. There is no catalytic subunit in CAF-1.

We have edited the text accordingly (p.11). We have replaced “catalytic” with “largest” to describe the CAF1A subunit, which is known to interact directly with PCNA.

(4) p.14: Only MCM2 has been shown to promote histone recycling at the replication fork.

We have edited the text accordingly (p.12).

(5) p.15: It should read “with diminished H3.3 occupancy in chromatin at the examined loci”.

We have edited the sentence accordingly (p.15).

(6) The authors should rephrase their conclusion about PIP-LA mutation not affecting the interaction between RECQ5 and RNAPII (p.18) as RECQ5-PIP-LA mutant pulls down more RNAPII than WT RECQ5 (Fig 7h).

We have edited the text accordingly and provided a possible explanation for this increase (p.19). We suggest that RNAPII α bound by the RECQ5-PIP-LA mutant was not dislodged from the chromatin due to the defective PCNA SUMO2 conjugation, resulting in the accumulation of the RNAPII α -RECQ5 mutant complex on the DNA.

(7) p.21: The following sentence “However, similar in vitro analysis suggests that PIAS1 is not the SUMO2 E3 ligase for PCNA” should refer to a publication or to data not shown.

We have clarified in the text that the statement is based on our own data that are not shown (p. 22).

Reviewer #3 (Remarks to the Author):

This paper describes the finding that SUMOylated PCNA – or at least a fusion of SUMO and PCNA is able to reduce replication-transcription conflicts.

It makes the case that PCNA is modified in transcribed regions in a manner that depends on interaction with RECQ5. SUMOylated PCNA in turn interacts with histone chaperones FACT and CAF1 encouraging H3.1 deposition and reduced RNAPolIII interaction. This activity reduces transcription dependent DNA damage.

The story is timely and a good fit for Nature Comms. I like the robust nature of the investigation and the clarity of the report. However in my opinion there are a couple of approaches that need to be tightened before publication.

The RECQ5-L mutant data is good, showing a link to PCNA SUMOylation, but it is not yet tied into the system in an entirely satisfactory way. Critically this is needed to confirm the RECQ5 link. Does complementation of RECQ5 depletion with this mutant result in transcription-mediated double-strand breaks, (slowed forks over transcribed regions?) and in RNA-poll lingering at fragile sites? Most significantly can its impact be over-come by expression of PCNA-SUMO2 (and not by PCNA-Ub)?

We thank the reviewer for this suggestion. In the revised manuscript, we have provided new data showing that transcription-mediated DNA breaks resulting from siRNA depletion of RECQ5 were suppressed by the overexpression of either S2-KR or CAF1, but not PCNA KR or Ub-KR, strongly supporting our conclusion that RECQ5 promotes SUMO2 modification to enhance the function of CAF1 to prevent transcription-mediated DNA breaks. These results are included in Fig. 7i-j and Suppl Fig. 9a-c.

While there are a great many questions this report raises I feel two are particularly out-standing. How does SUMO-PCNA improve FACT/CAF1A recruitment and how is RECQ5 responsible for SUMOylation of PCNA? How RECQ5 functions is clearly a story for another report but I feel a little more on the direct role of SUMOylated PCNA would flesh out the model considerably. With the model available this is relatively straight forward as mutations can be introduced into SUMO2 to prevent SIM interaction and/or to make the protein more SUMO1-like to address the role for a SIM from the PCNA point of view. Ideally expression of a SIM mutant of FACT/CAF1A would complement the approach.

CAF1A contains two PCNA-interacting peptide (PIP) motifs. Only the internal PIP2 is involved in CAF1-mediated nucleosome assembly (Rolef Ben-Shahar, MCB 2009), but PIP2 alone exhibits weak binding to PCNA. In addition to the PIP motifs, CAF1A also contains a SIM. Interestingly, the CAF1A SIM has been shown to preferentially interact with SUMO2/3, but not SUMO1, in vitro (Uwada, Biochem Biophys Res Commun 2010). We suggest that the simultaneous interactions between CAF1 PIP2 and PCNA and between CAF1 SIM and SUMO2 greatly improve and stabilize the associations between these two proteins. This is likely also true for SSRP1, which is a component of FACT that also contains a PIP variant and a SIM motif. Indeed, we have now included new data in Fig. 4d-f using CAF1A and SSRP1 SIM mutants to demonstrate that the SIM motifs of CAF1A and SSRP1 are involved in stabilizing the associations of CAF1 and FACT, respectively, with SUMO2-PCNA.

The idea that PIP-box + SIM makes a module for interacting with replication machinery in actively transcription regions is an exciting one. What is the proposed explanation for SUMO2, rather than SUMO1 involvement?

As mentioned above, the CAF1A SIM has been shown to interact with SUMO2/3, but not SUMO1, in vitro (Uwada, Biochem Biophys Res Commun 2010). This preference for SUMO2 binding may be the reason PCNA is specifically conjugated with SUMO2 to recruit CAF1. Cumulative studies indicate that

different SIMs may exhibit different preferences toward particular SUMO isoforms, although the mechanism for this differentiation still needs to be defined (Gareau & Lima 2010). While beyond the scope of this study, it is of great interest to use a structural approach to characterize the CAF1 and SSRP1 SIMs to understand their specificity toward the different SUMO isoforms. In addition, because we showed that PCNA-SUMO2 conjugation is induced by mechanisms distinct from those that induce SUMO1 conjugation, it is likely that distinct SUMO E3 ligases with different specificities toward various SUMO isoforms and responses to different cellular stimuli are required to conjugate SUMO1 and SUMO2 to PCNA. Alternatively, it is possible that the same SUMO E3 ligase can conjugate SUMO1 and SUMO2 onto PCNA in vitro, but in vivo, the specificity for SUMO2 conjugation is provided by a SUMO E3 ligase co-factor, such as RECQ5. We have included this discussion in the revised text (pp. 21-22).

I look forward to seeing the results of these experiments to finish this excellent report.

We thank the reviewer very much for the enthusiasm and support for our study.

REVIEWERS' COMMENTS:

Reviewer #1 (Remarks to the Author):

At this point the authors have addressed most of my concerns. The only issue that should be clarified is the authors' interpretation of a lack of change in R-loop levels. They suggest that this means PCNA does not resolve R-loops but rather replication-transcription collisions. However, in two recent papers (Hamperl et al., Lang et al., both in Cell) it was shown that collisions lead to R-loop formation. Therefore, it is anticipated that collision resolution factors should generally impact R-loop formation. The authors' should edit the text to address this seemingly contradictory interpretation in relation to the findings in the papers mentioned above. It is my opinion that once this issue is clarified, the paper should be accepted for publication in Nature Communications.

Reviewer #2 (Remarks to the Author):

The authors satisfactorily responded to the reviewers' comments. The revised manuscript is much improved and includes a number of previously missing controls so that the conclusions now more accurately reflect the authors' observations. Furthermore, the authors provide additional data. In particular, the rescue of the genome instability phenotype in RECQ5-depleted cells by CAF1 or SUMO2-PCNA expression and the characterization of SIM-dependent interactions of CAF-1 and FACT with PCNA constitute major additions to this work. Only a few minor points may require attention before this manuscript can be accepted for publication:

- The discussion about S2-PCNA vs. PCNA affinity for SIM mutants (end of p.13) is rather confusing and thus could be omitted.
- p.15: it should read: "elevated H3.1 levels were accompanied with diminished H3.3 occupancy in chromatin at some examined loci". Please remove "frequently" since this is not observed at all in HeLa cells.
- Fig. 5b: The authors demonstrate CAF-1 involvement in the chromatin assembly assay by using extracts depleted from CAF-1. They should include a western-blot showing CAF-1 depletion.
- Fig. 6d: the authors should indicate in the figure legend how many independent comet assays experiments were performed since only one representative experiment is shown.
- Fig.8 (model): it would be less confusing if interactions were represented as double arrows (RECQ5 with PCNA, CAF-1 and FACT with sumoylated PCNA). Note that non-sumoylated PCNA also binds CAF-1. SUMO2-PCNA could be indicated instead of SUMO-PCNA. It is not established if the pre-mRNA stays in place upon RNAPII release.

Reviewer #3 (Remarks to the Author):

The manuscript has been considerably improved and the questions I raised, particularly the mechanistic ones, answered.

REVIEWERS' COMMENTS:

Reviewer #1 (Remarks to the Author):

At this point the authors have addressed most of my concerns. The only issue that should be clarified is the authors' interpretation of a lack of change in R-loop levels. They suggest that this means PCNA does not resolve R-loops but rather replication-transcription collisions. However, in two recent papers (Hamperl et al., Lang et al., both in Cell) it was shown that collisions lead to R-loop formation. Therefore, it is anticipated that collision resolution factors should generally impact R-loop formation. The authors' should edit the text to address this seemingly contradictory interpretation in relation to the findings in the papers mentioned above. It is my opinion that once this issue is clarified, the paper should be accepted for publication in Nature Communications.

We have now included a short discussion on the lack of change in the R-loop level in the KR overexpressing cells (pp. 16-17). We suggest that even though KR overexpressing cells are prone to TRCs, which may lead to increasing R-loop production, these R-loop by-products are efficiently removed by multiple enzymes, including topoisomerases, RNase H1 and/or DNA helicases, such as BLM, all of which are expected to be functional in these cells.

Reviewer #2 (Remarks to the Author):

The authors satisfactorily responded to the reviewers' comments. The revised manuscript is much improved and includes a number of previously missing controls so that the conclusions now more accurately reflect the authors' observations. Furthermore, the authors provide additional data. In particular, the rescue of the genome instability phenotype in RECQ5-depleted cells by CAF1 or SUMO2-PCNA expression and the characterization of SIM-dependent interactions of CAF-1 and FACT with PCNA constitute major additions to this work.

We thank the reviewer for his/her support for our previous revision.

Only a few minor points may require attention before this manuscript can be accepted for publication:

- The discussion about S2-PCNA vs. PCNA affinity for SIM mutants (end of p.13) is rather confusing and thus could be omitted.

We agree with the reviewer that this discussion is confusing, and, therefore, we have removed this section (p.13).

- p.15: it should read: "elevated H3.1 levels were accompanied with diminished H3.3 occupancy in chromatin at some examined loci". Please remove "frequently" since this is not observed at all in HeLa cells.

We have removed "frequently" from the sentence (p.15).

- Fig. 5b: The authors demonstrate CAF-1 involvement in the chromatin assembly assay by using extracts depleted from CAF-1. They should include a western-blot showing CAF-1 depletion.

We have included a western blot in Supplementary Figure 6c to show that more than 80% of CAF1 was depleted from the extracts by CAF1 specific antibody.

- Fig. 6d: the authors should indicate in the figure legend how many independent comet assays experiments were performed since only one representative experiment is shown.

We have included this information in the corresponding figure legend (p.49).

- Fig.8 (model): it would be less confusing if interactions were represented as double arrows (RECQ5 with PCNA, CAF-1 and FACT with sumoylated PCNA). Note that non-sumoylated PCNA also binds CAF-1. SUMO2-PCNA could be indicated instead of SUMO-PCNA. It is not established if the pre-mRNA stays in place upon RNAPII release.

We have modified the cartoon model based the reviewer's suggestion. We changed the interactions to double arrows. We drew a dashed double arrow between CAF1 and non-modified PCNA. We changed the label to SUMO2-PCNA. We removed pre-mRNA from the drawing.

Reviewer #3 (Remarks to the Author):

The manuscript has been considerably improved and the questions I raised, particularly the mechanistic ones, answered.

We thank the reviewer for his/her support for our manuscript.

EDITORIAL REQUESTS:

There are a number of requirements that need to be addressed. We will be unable to proceed with acceptance of your manuscript until it adheres to these requirements. Please use the tracked changes feature of Microsoft Word to make these changes.

* *Nature Communications* uses a transparent peer review system, where for manuscripts submitted from January 2016 we are publishing the reviewer comments to the authors and author rebuttal letters of our research articles online as a supplementary peer review file. Please let us know in the cover letter when submitting the final version of your manuscript if you wish to opt out of this scheme or not. If you are concerned about the release of confidential data, we do permit redactions in the interest of confidentiality. If you would like to remove such information from these files, then please let us know specifically what information you would like to have removed. Please note that we cannot incorporate redactions for other reasons. For more information, please refer to our FAQ page at <https://media.nature.com/full/nature-assets/ncomms/authors/ncomms-transparent-peer-review.pdf>

We have indicated in our cover letter that we would like the journal to publish the reviewers' comments to our paper and our point-by-point responses.

* Your manuscript should comply with our policies and format requirements, detailed in our checklist for authors at:

http://www.nature.com/article-assets/npg/ncomms/authors/ncomms_manuscript_checklist.pdf

We have gone through the checklist to ensure that our manuscript complies with Nature Communications' policies and format requirements.

* **Data availability statements and data citations policy:** All *Nature Communications* manuscripts must include a section "Data Availability" at the end of the Methods section or main text (if no Methods). For more information on this policy, and a list of examples, please see <http://www.nature.com/authors/policies/data/data-availability-statements-data-citations.pdf>

- Accession codes for deposited data
- Other unique identifiers (such as DOIs and hyperlinks for any other datasets)
- At a minimum, a statement confirming that all relevant data are available from the authors
- If applicable, a statement regarding data available with restrictions
- If a dataset has a Digital Object Identifier (DOI) as its unique identifier, we strongly encourage including this in the Reference list and citing the dataset in the Data Availability Statement.

We have provided a Data Availability statement at the end of the Method section (p.35).

* **DATA SOURCES:** We strongly encourage authors to deposit all new data associated with the paper in a persistent repository where they can be freely and enduringly accessed. We recommend submitting

the data to discipline-specific, community-recognized repositories, where possible and a list of recommended repositories is provided here: <http://www.nature.com/sdata/policies/repositories>

If a community resource is unavailable, data can be submitted to generalist repositories such as figshare (<https://figshare.com/>) or Dryad Digital Repository (<http://datadryad.org/>). Please provide a unique identifier for the data (for example a DOI or a permanent URL) in the data availability statement, if possible. If the repository does not provide identifiers, we encourage authors to supply the search terms that will return the data. For data that have been obtained from publically available sources, please provide a URL and the specific data product name in the data availability statement. Data with a DOI should be further cited in the methods reference section.

Please refer to our data policies here: <http://www.nature.com/authors/policies/availability.html>

No molecular structure, microarray, nucleic acid or protein sequences were generated from this study.

* To ensure correct hyperlinking of the accession codes in your manuscript, please add the hyperlink or DOI in square brackets directly after the code throughout (for example, '5XRN [<http://dx.doi.org/10.2210/pdb5XRN/pdb>]', '1483958 [<https://dx.doi.org/10.5517/ccdc.csd.cc1t5m6>]', 'SRP109982 [<https://www.ncbi.nlm.nih.gov/sra/?term=SRP109982>]' or 'NQLW0000000 [https://www.ncbi.nlm.nih.gov/assembly/GCA_002312845.1/]').

Please pay particular attention to the following points, and use the tracked changes feature of Microsoft Word to make changes to the manuscript:

Not applicable to our manuscript.

* Please check whether your manuscript or Supplementary Information contain third-party images, such as figures from the literature, stock photos, clip art or commercial satellite and map data. We strongly discourage the use or adaptation of previously published images, but if this is unavoidable, please request the necessary rights documentation to re-use such material from the relevant copyright holders and return this to us when you submit your revised manuscript.

We confirm our manuscript and Supplementary information do not contain any third-party images.

* Please supply the main manuscript file in Microsoft Word or LaTeX format.

We have supplied the main manuscript in Microsoft Word with tracked changes.

* Please ensure all subheadings in the Results and Methods sections contain fewer than 60 characters including spaces. Please also ensure that these subheadings contain no punctuation.

We confirm that all subheadings fit the guideline.

* We strongly discourage the use of "data not shown" - in each case, please either supply the data as a supplementary figure, or if the statement is not important to the conclusions, remove it.

We have removed 'data not shown' and the corresponding sentence from the manuscript (p.22).

* In the Methods, please provide sufficient information such that the experiments could reasonably be reproduced without reference to other papers, and avoid use of the term 'as described previously'.

We have removed all 'as described previously' and replaced with sufficient experimental information.

* Please avoid using speech marks around words or phrases to convey emphasis.

We confirm no speech marks in the text.

* Please include a complete list of all primers used in your study, including primer names and sequences, in either the Methods section or Supplementary Information.

We have provided all the primers used in the study in either the Method section or in Supplementary Table 1 and 2.

* In the Methods, please ensure that the dilutions at which each antibody was used is stated, and catalogue numbers are provided for commercial antibodies.

We confirm that this information is provided in the Method section under "Antibodies" (pp.24-25).

* Please ensure that all blots and gels are accompanied by the locations of molecular weight/size markers. Blots should be cropped such that at least one marker position is present. Please also supply uncropped scans of the most important blots as a supplementary figure in the Supplementary

Information. This should be cited once in the Methods section.

We confirm that all blots are accompanied with molecular weight/size markers. All blots contained at least one molecular weight marker, and larger scan images of all the blots in the main figures are provided in Supplementary Figures 10-12.

*** Please ensure that +/- values are defined at the first point of use within the text and figure legends and numbers of replicates are given.**

We confirm that we have followed this guideline.

*** In each Figure and Supplementary Figure where error bars are used, they must be defined, and the number of experimental replicates stated. One statement at the end of each figure is sufficient if the error bars are equivalent throughout the figure.**

We confirm that we have followed this guideline.

*** Wherever p-values are stated in the main text and figure legends, please also state the name of the statistical test.**

We confirm that we have followed this guideline.

*** Where p-values are presented as symbols/letters, please ensure that all symbols/letters are defined in the relevant figure legend, together with the statistical test used.**

We confirm that all p-value presented as symbols are defined in the figure legends.

*** Please remove the scale bar labels from the figures in the main text and Supplementary Information (keeping the scale bar) and incorporate this information in the corresponding figure captions.**

We have removed scale bar labels from the figures and incorporated this information in the corresponding figure legends.

*** Please ensure that the corresponding dot plots are overlaid in the bar charts.**

We have replaced the bar chart in Figure 6d to a dot plot.

*** Please ensure that each display item is no larger than a pdf page size (260x179 mm).**

We confirm that the display items are no larger than a pdf page size.

*** Please ensure that figure legend titles are brief - they should not occupy more than one line in the final proof.**

We confirm that the figure legend titles fit this guideline.

*** Figures legends should not exceed 350 words, including titles. Please shorten by removing detailed methodological information and/or interpretation, or, if appropriate, consider splitting the affected figures in two. Note that we allow up to 10 display items (figures and tables) in the main manuscript.**

We confirm that all figure legends are within 350 words. We have total 9 displayed items (8 figures and 1 table) in the main manuscript.

*** Please ensure that figure panels are labelled using an a, b, c... convention.**

We confirm that all figure panels are labeled using a, b, c...convention.

*** Please define any new abbreviations, symbols or colours present in your figures in the associated legends, noting that these should be written out in words (blue circles, red dashed line, etc.) as symbols will not appear properly in the HTML text.**

We confirm that we have followed this guideline.

*** Please remove tables from the figures and supply as separate tables. Please note that tables may need to be renumbered so that they appear in numerical order in the main text.**

We have removed Figure 4a from Figure 4, and changed Figure 4a to Table 1, which can be found in the end of the main text.

*** Please ensure the references are in the standard Nature format and follow the sequence: author list, title of paper, name of journal, volume number, initial-final page numbers (year). Please note that dois**

are required only for online-only publications and correct journal abbreviations should be given.

We confirm that we have followed the standard Nature format for our references.

*** Please make a statement of competing financial and non-financial interests after the author contributions section that refers to all authors. If there are no competing interests, please add the statement "The authors declare no competing interests."**

We have provided the statement "The authors declare no competing interests" after the author contributions section (p. 35).

*** Please note that we do not reformat Supplementary Information files; they will be uploaded with the published article as they are submitted with the final version of your manuscript. Please check everything very carefully and remove any track changes from the file. Failure to adhere to our style guidelines will result in delays in production. The only sections we permit in the Supplementary Information file are Supplementary Figures, Supplementary Tables, Supplementary Methods, Supplementary Notes, Supplementary Discussion, Supplementary References.**

We confirm that we have checked all the Supplementary Figures and Data.

*** In the Supplementary Information file, please ensure that supplementary items are labelled and cited using only the following formats: Supplementary Figure 1, Supplementary Table 1, Supplementary Methods, Supplementary Note 1, Supplementary Discussion, Supplementary References. Please note the use of 'Supplementary' and that we do not use the 'S' prefix.**

We confirm that we have followed this format in the Supplementary Information file.

*** Throughout the main manuscript text, please use the following formats to cite Supplementary items: Supplementary Figure 1, Supplementary Table 1, Supplementary Methods, Supplementary Note 1, Supplementary Discussion, Supplementary References, Supplementary Movie 1, Supplementary Audio 1, Supplementary Data 1, Supplementary Software 1. Please note the use of 'Supplementary' and that we do not use the 'S' prefix.**

We confirm that we have followed this format in our main text.

*** We only permit one Supplementary Information file; please merge all Supplementary Information files into one PDF document. Only Supplementary Data/Software/Movie/Audio files should be submitted separately from the Supplementary Information.**

We confirm that we have provided all Supplementary figures and materials in one PDF document and have a separate Supplementary Data file for Mass Spec result (previous Supplementary Table 1).

*** Each Supplementary Figure should be accompanied by a legend, which should be presented below the figure and may be up to 350 words, that refers to all panels within the figure, and a title that summarises the figure and does not refer to specific panels. This also applies to spectra, which should be labelled as Supplementary Figures.**

We have placed a figure legend below each Supplementary Figure. We confirmed that all figure legends are within 350 words each.

*** Large datasets should be supplied as Supplementary Data files, whereas smaller tables should be supplied as Supplementary Tables. Please convert supplementary table 1 to a Supplementary Data file**

We have converted Supplementary Table 1 to Supplementary Data 1.

*** Please change panels in figure 8 to 'a' and 'b'.**

We have change panels in Figure 8 to 'a' and 'b'.

*** Nature journals require authors of life sciences research papers to include relevant details about several elements of experimental and analytical design in their manuscripts. This initiative aims to improve the transparency of reporting and the reproducibility of published results and is described at: <http://www.nature.com/authors/policies/reporting.pdf> To ensure that your manuscript complies with our policy, please pay close attention to the 'methods' and 'legends' sections of our checklist for authors: http://www.nature.com/article-assets/npg/ncomms/authors/ncomms_lifesciences_checklist.pdf You may also find the following collection of articles on statistics for biologists helpful: <http://www.nature.com/collections/qghhqm>**

We have gone through the checklist and confirm that the manuscript complies with the guideline.

*** Your paper will be accompanied by a two-sentence editor's summary, of between 250-300 characters, when it is published on our homepage. Could you please provide us with a suitably edited version.**

EDITOR'S SUMMARY

"Transcription-replication conflicts need to be resolved to minimise genome instability. Here the authors show that SUMO2 conjugated PCNA destabilizes RNAPII from chromatin, enhances replication progression and limits transcription-induced DNA damage at common fragile sites."

This summary is perfect!